# Chromosome-wide mechanisms to decouple gene expression from gene dose during sex-chromosome evolution

**Bayly S Wheeler[1†], Erika Anderson[1], Christian Frøkjær-Jensen[2,3‡], Qian Bian[1], Erik Jorgensen[2], Barbara J Meyer[1]***

[1]Department of Molecular and Cell Biology, Howard Hughes Medical Institute, University of California, Berkeley, Berkeley, United States; [2]Department of Biology, Howard Hughes Medical Institute, University of Utah, Salt Lake City, United States; [3]Danish National Research Foundation Centre for Cardiac Arrhythmia, University of Copenhagen, Copenhagen, Denmark

**Abstract** Changes in chromosome number impair fitness by disrupting the balance of gene expression. Here we analyze mechanisms to compensate for changes in gene dose that accompanied the evolution of sex chromosomes from autosomes. Using single-copy transgenes integrated throughout the *Caenorhabditis elegans* genome, we show that expression of all X-linked transgenes is balanced between XX hermaphrodites and XO males. However, proximity of a dosage compensation complex (DCC) binding site (*rex* site) is neither necessary to repress X-linked transgenes nor sufficient to repress transgenes on autosomes. Thus, X is broadly permissive for dosage compensation, and the DCC acts via a chromosome-wide mechanism to balance transcription between sexes. In contrast, no analogous X-chromosome-wide mechanism balances transcription between X and autosomes: expression of compensated hermaphrodite X-linked transgenes is half that of autosomal transgenes. Furthermore, our results argue against an X-chromosome dosage compensation model contingent upon *rex*-directed positioning of X relative to the nuclear periphery.

**\*For correspondence:** bjmeyer@berkeley.edu

**Present address:** [†]Department of Biology, Rhodes College, Memphis, United States; [‡]Department of Pathology, Stanford University, Stanford, United States

**Competing interests:** The authors declare that no competing interests exist.

## Introduction

Abnormalities in chromosome number (aneuploidy) have the potential to disrupt the balance of gene expression and thereby decrease organismal fitness and viability (*Siegel and Amon, 2012*). Aneuploidy occurs in most solid tumors and is a major cause of severe developmental defects and spontaneous abortions (*Siegel and Amon, 2012*). Unlike pathological imbalances in chromosome dose, the disparity in X-chromosome dose between 1X males and 2X females caused by sex-determination mechanisms has evolved to be well tolerated (*Charlesworth, 1996*). How this tolerance came about remains poorly understood. Of particular relevance is whether chromosome-wide regulatory mechanisms evolved to modulate the relationship between X-chromosome gene dose and gene product. Here we dissect the function and significance of gene regulatory strategies in the nematode *C. elegans* to achieve two goals: (1) elucidate mechanisms by which the X-chromosome dosage compensation process balances X expression between the sexes; (2) determine whether an X-chromosome-wide regulatory mechanism balances gene expression between X chromosomes and autosomes to facilitate X-chromosome evolution.

The need for X-chromosome-wide control of gene expression is illustrated by a description of sex-chromosome evolution (*Figure 1A*). For humans, although the X and Y sex chromosomes are genetically distinct, both originated from a single pair of homologous autosomes

**eLife digest** DNA inside cells is packaged into structures called chromosomes, each of which contains numerous genes. Many organisms, including humans, have two copies of most chromosomes in their cells. If the process of cell division goes awry, cells can end up with too many or too few copies of their chromosomes, which can cause serious illnesses.

Sex chromosomes pose a conundrum for cells. In humans, females have two copies of the X chromosome, whereas males only have one. This means that males have half the copy number (dose) of genes on the X chromosome. Human cells correct this imbalance by suppressing the activity, or expression, of most of the genes on one of the X chromosomes in females.

"Dosage compensation" also occurs in the roundworm species *Caenorhabditis elegans,* because male worms have one X chromosome whilst hermaphrodites have two. The dosage compensation mechanism in roundworms differs from that in humans. It involves turning down the expression of both hermaphrodite X chromosomes by half. The process is enacted by a dosage compensation complex that binds to specific sites along both hermaphrodite X chromosomes.

Dosage compensation mechanisms that reduce X chromosome expression in females cause sex chromosomes to have lower gene expression than non-sex chromosomes. Modern sex chromosomes evolved from a pair of non-sex chromosomes, and males lost one copy of all of the genes located on those ancestral chromosomes. This evolutionary history causes both sexes to have lower gene expression from X chromosomes than the other chromosomes, raising the question of whether a mechanism exists to balance out the difference in gene expression between sex chromosomes and non-sex chromosomes.

Wheeler et al. now show that the expression of any foreign gene artificially added to the X chromosomes of *C. elegans* is equalized between males and hermaphrodites despite the difference in gene dose. The equalization works regardless of where on the X chromosome the new gene is added. The foreign gene does not need to be adjacent to a binding site for the dosage compensation complex. These results indicate that dosage compensation mechanisms regulate gene expression on a chromosome-wide scale.

Wheeler et al. also show that genes added to X chromosomes are expressed at half the level as the same genes added to non-sex chromosomes. These results mean that no chromosome-wide mechanism balances gene expression levels between the X chromosome and the non-sex chromosomes.

It remains unknown how *C. elegans*, and many other living organisms, evolved to tolerate a lower level of gene expression from the sex chromosomes. Instead of a chromosome-wide mechanism, it is likely that individual genes evolved different ways to alter their expression levels. Working out what these mechanisms are remains a challenge for further research.

(*Charlesworth, 1996*). The differentiation of an autosome pair into two different sex chromosomes was proposed to begin when one homolog acquired a male-determining gene, thereby converting the homologs into a proto-Y and a proto-X (*Charlesworth and Charlesworth, 2000*; *Bachtrog, 2013*). As recombination ceased between the proto-Y and proto-X, the proto-Y could accumulate other male beneficial alleles, but the recombination isolation would cause it to degenerate into a gene-poor Y chromosome that did little more than specify male sexual fate. This process would result in males with one Y chromosome and one X chromosome and females with two X chromosomes. In the nematode *C. elegans*, sex chromosomes also likely arose from a pair of ancestral autosomes through a similar mechanism, but with Y-chromosome degradation progressing until the Y was lost completely (*Charlesworth, 1996*). The demise of Y as a male-determining chromosome was enabled by the emergence of a different sex-determination mechanism, one that utilized the ratio of X chromosomes to sets of autosomes (ploidy) to specify male (1X:2A) vs. hermaphrodite (2X:2A) sexual fates (*Nigon, 1951*).

The evolution of a male sex with only one X chromosome had the potential to impair male fitness. In the absence of any compensating mechanisms, genes present in one copy on the single male X would express half the level of gene products as genes present in two copies on the female X

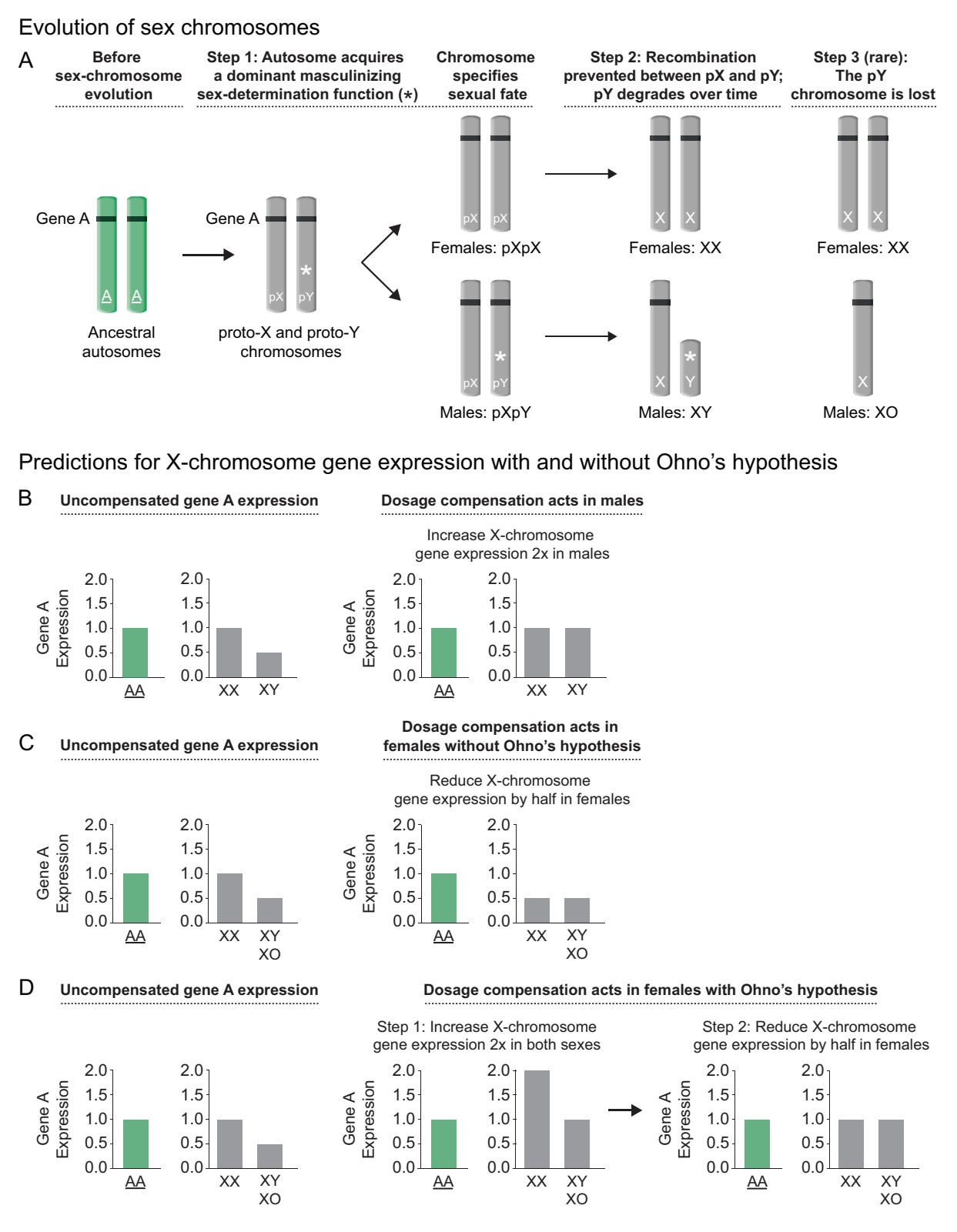

**Figure 1.** Sex chromosome evolution and its impact on gene expression. (A) Sex chromosome evolution. In mammals, the X and Y sex chromosomes were derived from a single pair of homologous autosomes referred to as the ancestral autosomes (green). Before the evolution of sex chromosomes, genes represented by gene A (black) were present on both ancestral autosomes (AA). During sex chromosome formation, one autosome acquired a dominant male-determining gene (*), thereby converting an ordinary autosome pair into a proto-X (pX) chromosome and a male sex-determining

*Figure 1 continued on next page*

*Figure 1 continued*

proto-Y (pY) chromosome (step 1). As recombination ceased between the proto-Y and proto-X, and the proto-Y accumulated other male beneficial alleles, the proto-Y degenerated into the present-day gene-poor Y chromosome that specified male fate (step 2). Loss of genes from Y (e.g. gene A) caused genes from the ancestral autosome to be present in only one copy in males instead of two copies on the ancestral autosomal pair. While most mammalian sex chromosomes progressed only through steps 1 and 2, the nematode sex chromosomes were proposed to have evolved by a similar route but then to have undergone an additional step in which Y chromosome degradation progressed until the Y was lost completely (step 3), giving rise to XX hermaphrodites and XO males. Demise of Y was enabled by the emergence of a sex-determining mechanism that utilizes the ratio of X chromosomes to sets of autosomes (X:A signal) to determine sex rather than the dominant masculinizing gene that initiated sex-chromosome evolution. (B, C, D) Predictions for X-chromosome gene expression with and without Ohno's upregulation mechanism (B) Prediction for X-linked gene expression when the dosage compensation mechanism increases expression of X in males, a case not requiring Ohno's hypothesis. If the dose-sensitive gene A were expressed at a level of 1 when present in two copies on the ancestral autosomes, it would be expressed at a level of 0.5 in present-day males with only one copy on the single male X, and at a level of 1.0 in present-day females, which carry one copy on both X chromosomes. If gene A were haploinsufficient, its reduced expression could have deleterious consequences for the male. To compensate for reduced gene expression in males, a dosage compensation mechanism arose to balance X expression between the sexes. *Drosophila melanogaster* increases X-linked gene expression two-fold in males, thereby balancing the level of gene expression with that in present-day females and that in the ancestral species prior to sex-chromosome evolution. (C) Prediction for X-linked gene expression when the dosage compensation mechanism reduces X gene expression in females without an accompanying upregulation mechanism proposed by Ohno. X-chromosome dosage compensation in mammals and *C. elegans* occurs by mechanisms different from that of Drosophila, even though sex chromosomes may have evolved by a similar route. These species compensate for the imbalance in X-chromosome dose between the sexes by reducing X-linked gene expression in females/hermaphrodites by half, causing both sexes to express gene A at half the level of the ancestral species prior to the evolution of sex chromosomes. (D) Prediction for X-linked gene expression when Ohno's mechanism of upregulation operates and the dosage compensation mechanism reduces X gene expression in females. Recognizing that reducing X-chromosome gene expression in females as a mechanism of dosage compensation between sexes might create a deleterious reduction in X-chromosome products for both sexes, Susumo Ohno proposed a two-step mechanism for the regulation of X gene expression. After the degeneration of Y began but before the evolution of dosage compensation, a mechanism would arise to increase X-chromosome gene expression two-fold in both sexes (step 1). This upregulation of X expression would make expression from the male X equal to that of the ancestral autosomes but would cause a two-fold overexpression of X-linked genes in females relative to the ancestral autosomes. The overexpression in females would then be offset by an X-chromosome dosage compensation process that reduced X expression in females, thereby balancing X expression between males and females, as well as balancing expression between female X chromosomes and the ancestral autosomes (step 2).

The following figure supplement is available for figure 1:

**Figure supplement 1.** Average gene expression levels vary across autosomes.

chromosomes and two copies on the ancestral autosomes. Reduced expression of dose-sensitive genes on X would likely decrease male viability (*Figure 1A*).

For the XX/XY species *Drosophila melanogaster,* potential complications caused by males having one X chromosome were averted by the co-evolution of a dosage compensation mechanism that operates by doubling gene expression from the single male X chromosome. Elevating X expression in males balanced gene expression with that from the two female X chromosomes, while maintaining similar expression between the single X and the two ancestral autosomes (*Figure 1B*) (*Lucchesi and Kuroda, 2015*). In contrast, other animals including placental mammals and the nematode *C. elegans* developed mechanisms of dosage compensation that equalized X-chromosome expression between the sexes by reducing X-linked gene expression by half in XX animals (*Figure 1C*) (*Meyer, 2010*; *Galupa and Heard, 2015*). Although decreasing X-chromosome gene expression in XX animals balances X expression between the sexes, it causes females to have the same problem as males: insufficient levels of X-chromosome products relative to those of ancestral autosomes (*Figure 1C*).

For mammals, Susumo Ohno proposed that loss of genes from the degenerating Y chromosome was compensated via two sequential evolutionary steps (*Figure 1D*) (*Ohno, 1967*). In the first step, mechanisms would arise to increase expression of each X-linked gene in both sexes by approximately two fold. While this increase in gene expression would offset the X-chromosome dose deficiency in males, it would cause overexpression of X-linked genes in females. In a second step, overexpression of genes on female X chromosomes would be offset by inactivating one of the two X chromosomes. Controversy has surrounded the question of whether mammals and other organisms such as *C. elegans,* which reduce female X-chromosome gene expression to equalize X expression between the sexes, do indeed employ a separate compensating mechanism to increase gene

expression on X chromosomes of both sexes (*Xiong et al., 2010*; *Deng et al., 2011*; *Kruesi et al., 2013*; *Albritton et al., 2014*).

The controversy in mammals caused Ohno's hypothesis to be re-evaluated by a multi-species approach (*Julien et al., 2012*). The central prediction of Ohno's two-step hypothesis is that the single active X chromosome of males and females will be expressed at the same level as the combined ancestral autosomes. Although expression of ancestral autosomes cannot be measured directly for mammals, it can be estimated, because mammalian sex-chromosome evolution occurred after the divergence of birds and mammals. Thus, expression of orthologous autosomal genes in chickens serves as an estimate for expression of genes on the mammalian proto-X. Comparison between the extant mammalian X chromosome and the orthologous chicken autosome failed to reveal evidence for X-chromosome-wide upregulation in placental mammals (*Julien et al., 2012*). In these species, genes on the single active X chromosome in males and females are expressed, on average, at half the level of the orthologous pair of autosomes, contrary to Ohno's hypothesis. Although the experimental approach failed to identify a chromosome-wide transcriptional mechanism of X upregulation, it left open the possibility that regulatory mechanisms might have arisen on a gene-by-gene basis to compensate for low activity of critical X-linked genes caused by chromosome-wide reduction of X expression. In contrast, evidence in favor of Ohno's hypothesis exists in marsupials, suggesting that X-chromosome upregulation may have accompanied sex-chromosome evolution in some lineages but not others (*Julien et al., 2012*).

For *C. elegans*, tests of Ohno's upregulation hypothesis have faced two major obstacles. First, limited information about the orthology of nematode genes relative to other species makes it premature to estimate the level of gene expression for the *C. elegans* proto-X chromosome from the expression level of autosomal orthologs in other species. In the absence of information about orthology, the assumption was made in some studies that the average overall expression of all genes on extant autosomes would serve as a proxy for expression of the proto-X chromosome (*Deng et al., 2011*; *Kruesi et al., 2013*). However, this assumption is undermined by the unexpected observation we report here that average gene expression varies widely (1.9 fold) among the five different autosomes (*Figure 1—figure supplement 1A–D*). Hence the previous finding that X expression is equivalent to the average level of autosomal expression does not confirm Ohno's hypothesis. Second, X-chromosome gene expression undergoes transcriptional silencing in germ cells of XX and XO animals (*Figure 1—figure supplement 1E*) (*Reinke et al., 2000*; *Kelly et al., 2002*; *Deng et al., 2011*; *Gaydos et al., 2012*), which comprise 68% of all adult cells (*Crittenden et al., 2006*; *Morgan et al., 2010*), causing tests of Ohno's hypothesis that quantify adult gene expression (*Xiong et al., 2010*) to underestimate X expression by about 50%. Thus, no compelling evidence supports or refutes Ohno's hypothesis for *C. elegans*.

Here in a different exploration of Ohno's hypothesis, we asked whether a chromosome-wide mechanism operates in *C. elegans* to upregulate X-linked gene expression in both sexes. Our approach quantified gene expression specifically in somatic cells and did not rely on untested assumptions about expression levels of ancestral autosomes. We quantified expression in L1 larvae of the same transgenes integrated in single copy on either the X chromosome or autosomes. The L1 developmental stage occurs prior to the onset of germline proliferation, thereby preventing germline silencing from interfering with our quantification of X-chromosome expression. Moreover, monitoring expression of the same gene in the same species while varying only its location within the genome enabled a direct comparison of X and autosomal expression levels that tests one attractive molecular mechanism (a chromosome-wide mechanism) for upregulating X-chromosome transcription to balance gene expression between X and autosomes.

Our transgene approach also enabled us to determine whether the dosage compensation process, which equalizes X expression between the sexes, acts chromosome-wide to control gene expression all along X or instead acts locally on a gene-by-gene basis. In *C. elegans*, as in mammals, not all genes on X are dosage compensated (*Carrel and Willard, 2005*; *Jans et al., 2009*; *Kruesi et al., 2013*), and the factors that determine whether a gene becomes dosage compensated or escapes from dosage compensation are not known. In particular, it has been difficult to tease apart whether a gene's local DNA sequence, its proximity to a binding site for the dosage compensation machinery, its position on the chromosome, its location within the nucleus, or a combination of such factors influences the dosage compensation process. By monitoring the expression of identical transgenes integrated at various locations along the X chromosome and autosomes, with and

without a co-integrated binding site for the dosage compensation machinery, we eliminate the contribution of gene-specific differences in DNA sequence on gene expression and assess the role of chromosome location and proximity to a binding site on the regulation of X-chromosome gene expression, thereby differentiating a global chromosome-wide process from a local gene-by-gene process.

Balancing X-chromosome gene expression between the sexes is achieved in *C. elegans* by a dosage compensation complex (DCC) that is homologous to condensin (*Csankovszki et al., 2009*; *Mets and Meyer, 2009*; *Meyer, 2010*), a conserved protein complex that controls the compaction and resolution of all mitotic and meiotic chromosomes prior to their segregation (*Wood et al., 2010*; *Hirano, 2016*). The DCC is recruited to both X chromosomes of hermaphrodites by *cis*-acting regulatory elements distributed throughout X called r̲ecruitment e̲lements on X̲ (*rex* sites). These sites include DNA motifs that are highly enriched on X chromosomes and important for DCC binding (*Csankovszki et al., 2004*; *McDonel et al., 2006*; *Ercan et al., 2007*; *Jans et al., 2009*; *Pferdehirt et al., 2011*). Once bound to X, the DCC remodels the topology of X, while reducing the expression from both hermaphrodite X chromosomes by half to balance gene expression between the sexes (*Crane et al., 2015*).

We first show here that all transgenes integrated onto *C. elegans* X chromosomes are dosage compensated, regardless of their position on X and hence their proximity to an endogenous *rex* site. Thus, the X chromosome is broadly permissive for the transcriptional repression that achieves dosage compensation. Furthermore, integration of the same transgenes onto autosomes, either with or without an adjacent DCC-bound *rex* site, failed to elicit DCC-mediated repression in hermaphrodites. Thus, DCC binding to a nearby *rex* site is not sufficient to trigger dosage compensation of a gene, nor is it necessary. These data reinforce a model of dosage compensation in which the DCC acts through multiple *rex* sites to induce chromosome-wide changes in X structure that influence expression of endogenous and engineered genes (*Crane et al., 2015*).

While our transgene approach demonstrates a robust chromosome-wide mechanism to balance X gene expression between the sexes, it provides strong evidence against an analogous, chromosome-wide mechanism that would fulfill Ohno's hypothesis for balancing gene expression between X chromosomes and autosomes. We show that in dosage-compensated (i.e. down regulated) XX animals, the per-copy expression of X-linked transgenes is half, not equivalent to, the per-copy expression of their counterparts on autosomes. In addition, the per-copy expression of hemizygous X-linked transgenes in XO animals is equivalent to, not double, the per-copy expression of their autosomal counterparts. Both findings are inconsistent with a chromosome-wide mechanism of upregulation. Our results suggest that if upregulation did occur to compensate for gradual loss of genes during X-chromosome evolution, it proceeded by the emergence of diverse gene-specific mechanisms that would compensate for their reduced dose.

Finally, our analysis of X-chromosome regulation, combined with chromosome localization studies, allowed us to evaluate a recent, speculative model of X-chromosome dosage compensation, which proposes that *rex* sites target X chromosomes to the nuclear periphery in males to increase gene expression, while DCC binding to *rex* sites in hermaphrodites relocates X to the interior, thereby reducing gene expression to achieve dosage compensation (*Sharma et al., 2014*; *Sharma and Meister, 2015*). Results presented here provide strong evidence against this model of dosage compensation. Together, our studies offer key insights into mechanisms by which abnormalities in chromosome number can evolve to be well tolerated.

## Results

### Expression of transgenes integrated across X is balanced between the sexes by the condensin-driven dosage compensation process

To analyze mechanisms that regulate gene expression across X, we examined the expression of 28 reporter genes integrated at 12 different sites on X and 36 reporter genes integrated at 14 different sites dispersed among the five autosomes (*Figure 2A*). Single-copy transgene cassettes, each containing two reporters, were integrated into the genome using either targeted or random Mos1-mediated insertion (*Frøkjær-Jensen et al., 2008*, *2014*). Cassettes included both *Cbr-unc-119*, a neuronally expressed gene from the sister Caenorhabditid *C. briggsae*, and a fluorescent reporter

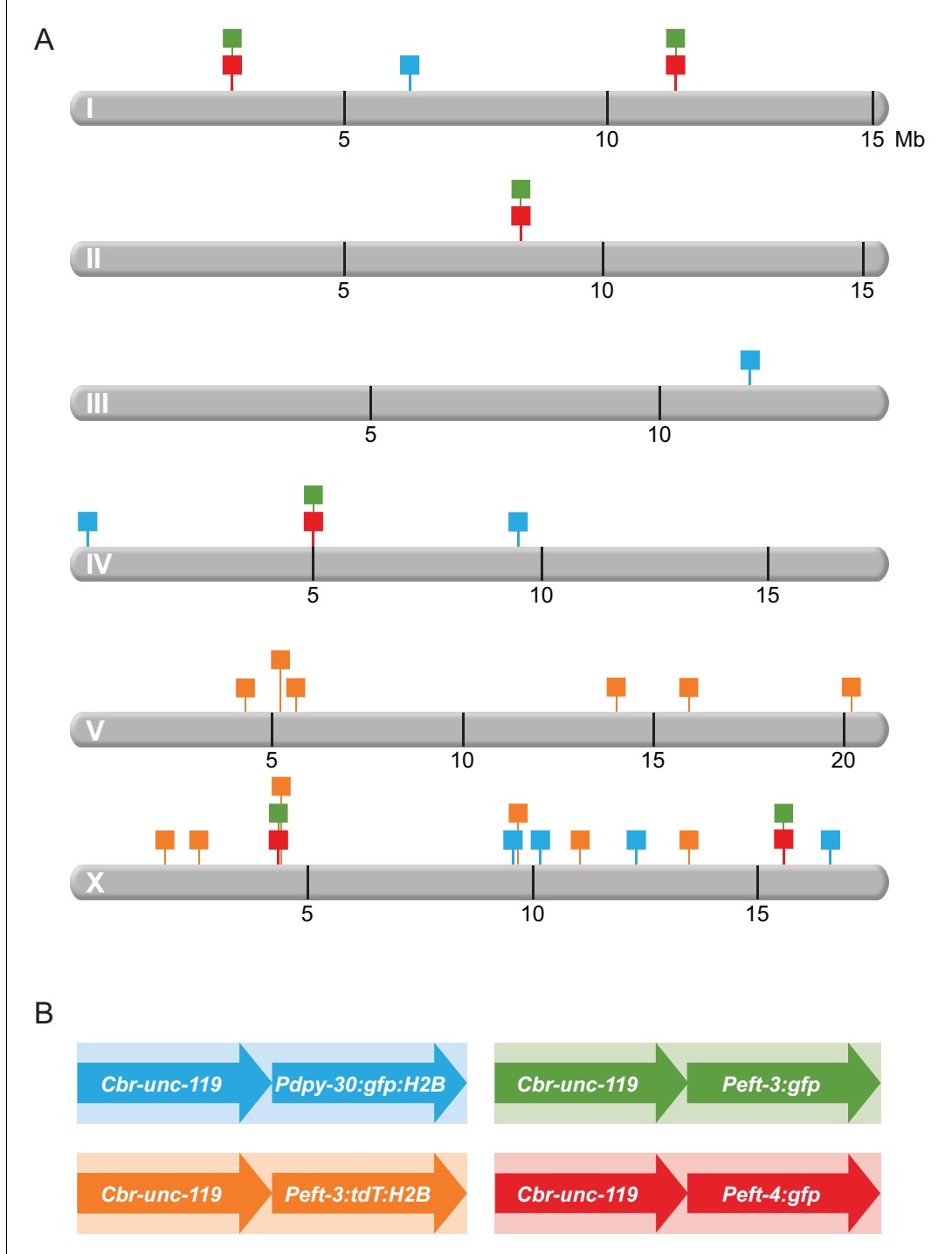

**Figure 2.** Schematic diagram of transgenes integrated throughout the genome. (A) Chromosomal locations of transgene cassettes are represented by square flags marking the insertion sites on all six *C. elegans* chromosomes. (B) A transgene cassette is composed of two distinct reporters integrated at each site: *Cbr-unc-119* and a fluorescent reporter with one of three *C. elegans* promoters. The flag color in (A) corresponds to the composition of the cassettes (shown in B. Transgene cassettes containing *Peft-3:gfp* (green) and *Peft-4:gfp* (red) were inserted in the same four sites using targeted Mos1-mediated Single Copy Insertion (mosSCI). Transgene cassettes containing *Pdpy-30:gfp:H2B* (blue) and *Peft-3:tdTomato:H2B* (orange) were inserted randomly throughout the genome using miniMos. Cassettes of each type are numbered sequentially, from left to right, along a chromosome. For example, in subsequent figures, the first 'green cassette' on the left end of chromosome I will be indicated by a green flag and 'Chr I, site 1'; the second 'green cassette' will be 'Chr 1, site 2'. The first 'green cassette' on the left end of X will be indicated by a green flag and 'Chr X, site1'.

(*gfp* alone, *gfp* fused to histone H2B, or *tdTomato* fused to histone H2B) driven by the promoter of a ubiquitously expressed *C. elegans* gene (*dpy-30, eft-3*, or *eft-4*) (*Figure 2B*). *dpy-30* is an essential autosomal gene that acts independently in the DCC and the COMPASS complex, which makes the active chromatin modification H3K4me3 (*Hsu and Meyer, 1994*; *Miller et al., 2001*; *Nagy et al., 2002*; *Pferdehirt et al., 2011*; *Hsu et al., 1995*). *eft-3* (autosomal, also called *eef-1A.1*) and *eft-4* (X-linked, also called *eef-1A.2*) encode essential paralogous translation elongation factors (*Maciejowski et al., 2005*). Use of multiple promoters and reporters with different expression levels and tissue specificities allowed us to test diverse gene regulatory scenarios for responsiveness to dosage compensation.

Reporter gene expression was quantified from populations of L1 larvae that had been synchronized to within three hours of hatching. This strategy conferred two advantages. It eliminated any confounding influence of X-chromosome silencing in germ cells, since the L1 stage of development occurs before the onset of germline proliferation. It also minimized gene expression differences due solely to the activation or repression of genetic pathways operating at different developmental stages.

To determine whether transgenes integrated on X are regulated by the DCC, we compared the overall gene expression levels in homozygous wild-type XX animals (2 copies of transgenes), homozygous dosage-compensation-defective XX animals (2 copies of transgenes), and hemizygous wild-type XO animals (1 copy of transgenes) using quantitative reverse-transcriptase PCR (qRT-PCR). To be considered dosage compensated, a transgene should have increased expression in DCC-defective XX animals compared to control XX animals, and it should have the same overall level of expression from the single copy in XO males as the two copies in XX hermaphrodites. That is, the single transgene copy in the male should be expressed at twice the level as either of the two copies in the wild-type hermaphrodite. To disrupt dosage compensation, we used RNAi to deplete SDC-2, the sole hermaphrodite-specific DCC subunit that triggers assembly of DCC subunits onto X (*Dawes et al., 1999*). *sdc-2(RNAi)* causes overexpression of X-linked genes and XX-specific lethality (*Nusbaum and Meyer, 1989*).

Depletion of SDC-2 activity not only increases expression of X-linked genes, it mildly reduces expression of about 30% of autosomal genes (*Jans et al., 2009*; *Kruesi et al., 2013*) (*Figure 1—figure supplement 1F*), making it essential to identify autosomal genes not affected by *sdc-2(RNAi)* for use in normalizing gene expression. For normalization candidates, we selected 12 autosomal genes that had similar expression levels between control and *sdc-2(RNAi)* animals, as assayed by GRO-seq, microarray, and RNA-seq experiments (*Jans et al., 2009*; *Kruesi et al., 2013*). We then followed the geNorm approach (*Vandesompele et al., 2002*) to identify the three most stably expressed autosomal genes (*cdc-42, H06O01.1*, and *Y38A10A.5*) from three replicates of control and *sdc-2(RNAi)* animals.

To verify that our normalization approach recapitulated RNA-seq data, we quantified gene expression of two dosage compensated genes on X (*F41E7.5* and *F47B10.2*), one non-compensated gene on X (*C15C7.5*), and one autosomal gene (*F19F10.91*) in both control and *sdc-2(RNAi)* worms by qRT-PCR. The two DCC-regulated genes were significantly upregulated in *sdc-2(RNAi)* L1s compared to control L1s, and both the autosomal gene and the non-dosage-compensated X gene were not significantly affected by *sdc-2(RNAi)* (*Figure 3—figure supplement 1A*), thus validating our qRT-PCR approach for assessing the dosage compensation status of any gene.

Quantification of reporter mRNA levels to assess whether transgenes integrated across X were dosage compensated revealed that all 28 X-linked reporters were repressed by the DCC. All had increased expression in *sdc-2(RNAi)* versus control L1s, regardless of their location on X and the origin of their promoter, whether from the *C. elegans* X chromosome (*eft-4*), *C. elegans* autosomes (*eft-3* and *dpy-30*), or a *C. briggsae* autosome (*Cbr-unc-119*) (*Figure 3A*). The extent of DCC-mediated repression ranged from 1.3 to 3.7-fold, consistent with the range observed for endogenous dosage-compensated X-linked genes assessed by RNA-seq (*Figure 1—figure supplement 1F*).

We also found that the X-linked reporters had equivalent expression in XX and XO animals at the L1/L2 stage. That is, the total level of transgene expression from the two X chromosomes of hermaphrodites was not statistically different from the total level of transgene expression from the single X of males (*Figure 3B*). Together these data show that the dosage compensation process creates a chromosome-wide environment that permits repression of transgenes integrated all along the X chromosome, resulting in equivalent transcription between the sexes.

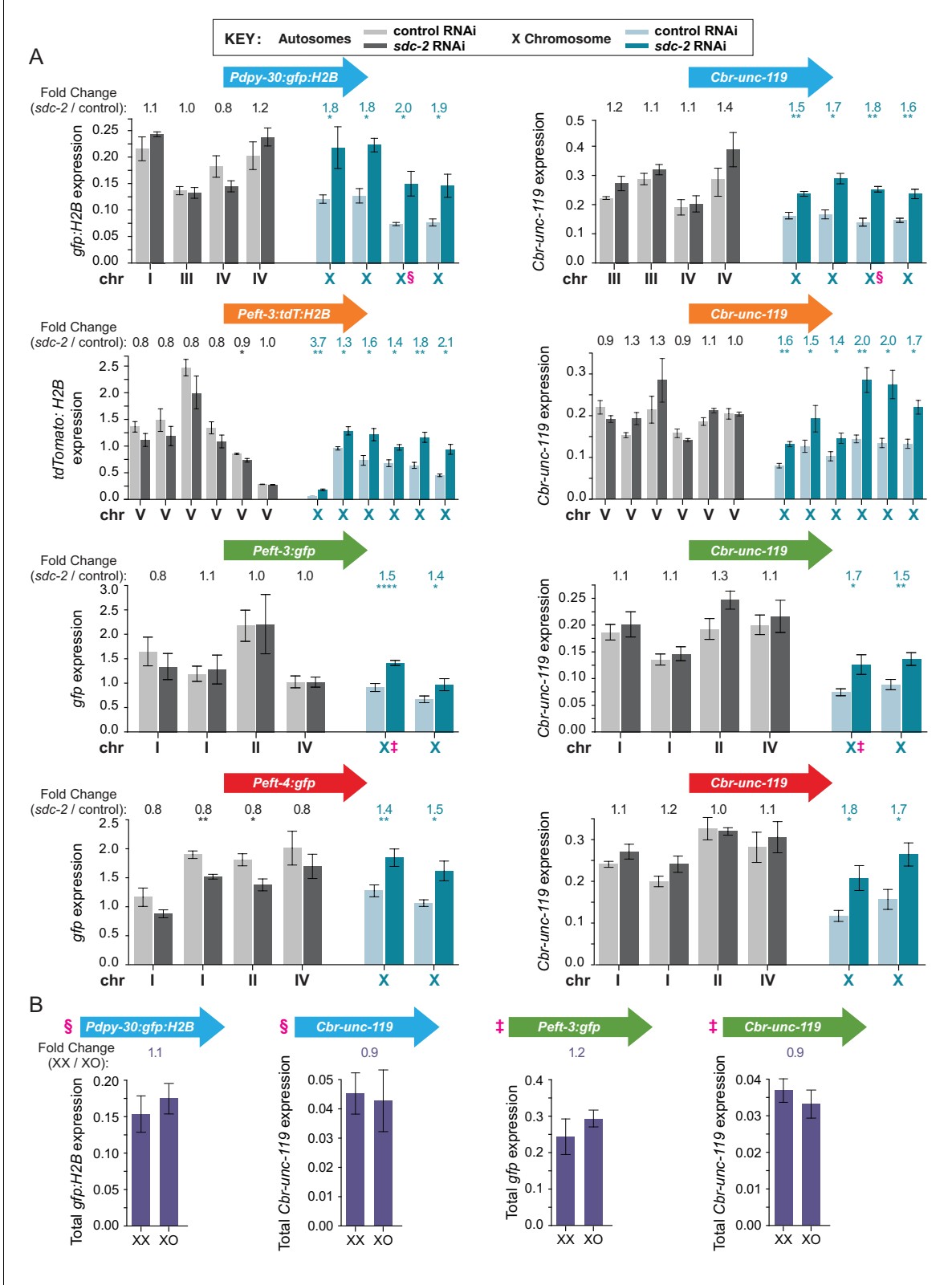

**Figure 3.** Transgenes integrated on X but not autosomes are regulated by the DCC, which balances X expression between sexes. (**A**) Quantification in control RNAi XX (light) or *sdc-2(RNAi)* XX (dark) L1 larvae of mRNA levels for the two reporters in each transgene cassette on X chromosomes (blue) and autosomes (gray). The bars represent the average level of expression among at least three biological replicates for each reporter in a cassette. Data for reporters are presented in the same order, from left to right, as the order of transgene cassettes along a chromosome (see *Figure 2*). The fold change

*Figure 3 continued on next page*

Wheeler *et al.* eLife 2016;5:e17365. DOI: 10.7554/eLife.17365

*Figure 3 continued*

in gene expression between *sdc-2(RNAi)* XX (dark) and control XX animals (light) is shown above each transgene, with the number asterisks indicating the p-value: p≤0.05, one asterisk; p≤0.01, two asterisks; p≤0.0001, four asterisks (Student's t-test). Error bars show the standard error of the mean for at least three biological replicates. All reporters on X show significant elevation in gene expression in the dosage-compensation-defective *sdc-2(RNAi)* XX L1s compared to control XX L1s. In contrast, none of the reporters on autosomes exhibit a significant increase in expression in *sdc-2(RNAi)* XX L1s. A few autosomal reporters (see *Peft-3:tdT:H2B* and *Peft-4:gfp*) exhibit slight but significant reduction in gene expression, consistent with previous genome-wide measurements of gene expression in dosage compensation mutants (*Jans et al., 2009*; *Kruesi et al., 2013*; *Crane et al., 2015*). (B) Comparison of mRNA levels in XX vs. XO L1/L2 animals for transgene cassettes integrated on X. Total reporter mRNA levels were quantified in XX animals that were homozygous for the transgene cassettes (2 copies of each reporter) and XO animals that were hemizygous for the transgene cassette (1 copy of each reporter). Reporters in cassettes selected for this experiment are designated by § or ‡ in panels A and B. The fold change in gene expression is indicated above each pair of measurements in XX and XO animals. The expression levels were not statistically different between the two copies in XX animals vs. the single copy in XO animals, indicating that the reporters in each transgene cassette became dosage compensated (Student's t-test). Error bars show the standard error of the mean for at least three biological replicates.

The following figure supplement is available for figure 3:

**Figure supplement 1.** Transgene expression levels are highly consistent for sites across the X chromosome and for sites across an autosome in XX animals.

In contrast to transgenes on X, expression of 36 transgenes on autosomes was not increased by disrupting dosage compensation (*Figure 3A*). One *Peft-3:tdTomato:H2B* autosomal reporter and two *Peft-4:gfp* autosomal reporters had a slight but significant decrease in gene expression upon *sdc-2* depletion FC (Fold Change) = 0.86, p=0.03; FC = 0.76, p=0.03; FC = 0.8, p=0.008). These decreases were consistent with the effect of *sdc-2* mutations on 30% of autosomal genes in prior genome-wide experiments (*Jans et al., 2009*; *Kruesi et al., 2013*).

## Dosage compensation of transgenes on X does not require a DCC binding site nearby

Prior genome-wide studies found that DCC binding near an endogenous X-linked gene was neither necessary nor sufficient for the dosage compensation of the gene (*Jans et al., 2009*; *Kruesi et al., 2013*). We re-examined this issue using the transgenes. The DCC is recruited to endogenous X chromosomes by sequence-dependent recruitment elements on X (*rex* sites) and spreads to lower-affinity bindings sites, called *dox* sites (dependent on X), located in promoters of actively transcribed genes (*Csankovszki et al., 2004*; *McDonel et al., 2006*; *Jans et al., 2009*; *Pferdehirt et al., 2011*). DCC occupancy at *dox* sites correlates directly with the expression level of the gene (*Jans et al., 2009*; *Pferdehirt et al., 2011*). To assess the relationship between transgene repression and proximity to a DCC binding site, we used chromatin immunoprecipitation (ChIP) to quantify levels of the DCC components DPY-27 (an SMC condensin subunit) and SDC-3 (a zinc finger protein required for condensin loading) bound at *Peft-3:gfp* and *Cbr-unc-119* on X. Binding of both DPY-27 and SDC-3 to the transgenes was negligible compared to DCC binding at the strong *rex* sites *rex-1* and *rex-32* (*Figure 4A*). The finding that transgenes lacking DCC binding are nonetheless repressed by the DCC indicates that local DCC binding is not required for dosage compensation.

We then asked whether a transgene's distance from a *rex* site is correlated with its ability to undergo DCC-mediated repression (gene expression fold change in *sdc-2(RNAi)* vs. control worms) (*Figure 4B–D*). We found that for transgenes on X, the increase in expression in *sdc-2(RNAi)* animals was not correlated with their proximity to a *rex* site. Thus, a nearby *rex* site is not essential for the compensation of a gene. Furthermore, DCC binding to a nearby *dox* site is also not essential for the compensation of a gene (*Figure 4—figure supplement 1*). Together, these data indicate that the DCC can act at a distance to control expression of foreign genes integrated across the X chromosome.

## A closely linked *rex* site with high DCC occupancy is not sufficient to elicit repression in XX animals of transgenes integrated on autosomes

The finding that nearby DCC binding is not necessary for a transgene on X to become dosage compensated caused us to ask whether a closely linked *rex* site could elicit DCC-mediated repression of

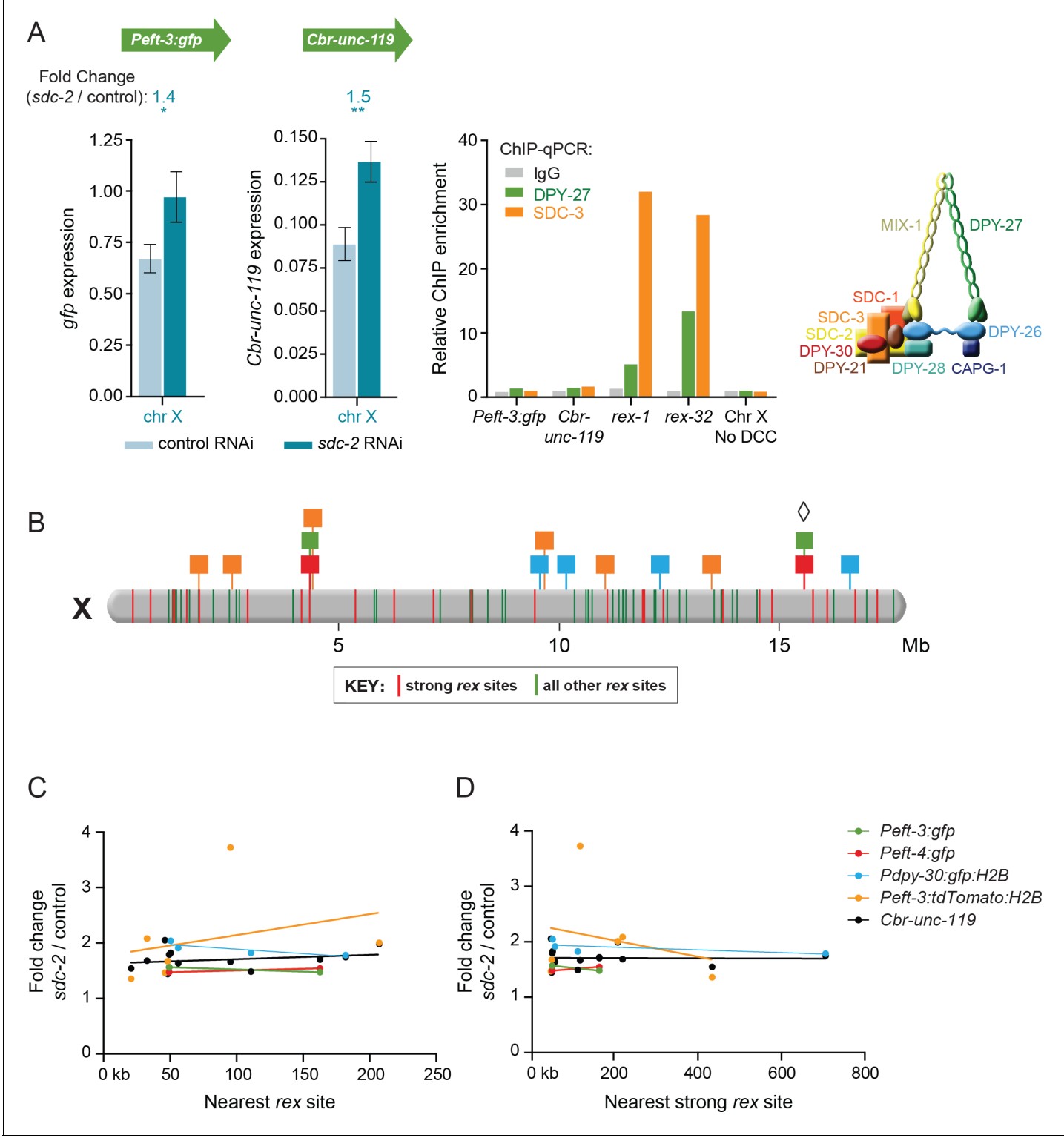

**Figure 4.** Dosage compensation of transgenes does not require local DCC binding. (**A**) Quantification of mRNA levels and DCC binding for the *Peft-3: gfp* and *Cbr-unc-119* reporters integrated at position 15.6 Mb on chromosome X. Gene expression is represented as in *Figure 3*. ChIP was conducted using antibodies against DCC subunits DPY-27 (green) or SDC-3 (orange), and the negative control IgG (gray). ChIP enrichment was calculated using quantitative PCR with primers for the *Peft-3:gfp* and *Cbr-unc-119* reporters, the strong *rex* sites *rex-1* or *rex-32*, and a region of X that does not recruit the DCC. Enrichment is expressed relative to an autosomal region that does not recruit the DCC and is normalized to input. A schematic diagram of the DCC is shown. The two reporters became dosage compensated even though no DCC complex was detected at either reporter in the integrated

*Figure 4 continued on next page*

*Figure 4 continued*

transgene cassette. (**B**) A schematic diagram of relative positions for X-linked transgene cassettes and endogenous *rex* sites on the X chromosome. Cassette locations are represented as flags, as in *Figure 1*. Colored bars on X indicate the positions and strength of endogenous *rex* sites. The 25 strongest *rex* sites are shown in red; all other rex sites are shown in green. *rex*-site strength was assessed by SDC-3 occupancy in ChIP-seq experiments. The diamond indicates the *Peft-3:gfp* and *Cbr-unc-119* transgene cassette tested for DCC binding in part A. (**C** and **D**) Scatter plots compare the fold change in gene expression of reporters vs. the distance of the nearest *rex* site of any strength (**C**) or the nearest strong *rex* site (**D**). No correlation was found between the extent of a reporter's increase in expression in DCC-defective animals and its proximity to a *rex* site. Linear regression lines are shown for each transgene.

The following figure supplement is available for figure 4:

**Figure supplement 1.** Transgenes do not require close proximity of a *dox* site to become dosage compensated.

---

transgenes on autosomes in XX animals. We analyzed expression of *Cbr-unc-119* and *Peft-3:gfp* in transgene cassettes that were integrated with and without *rex-32* at four sites on autosomes.

We first showed the DCC binds to these ectopic *rex* sites. SDC-3 binding at the ectopic *rex-32* sites on autosomes was similar to, or greater than, binding at the endogenous *rex-32* site on X (*Figure 5B and C*). Similarly, DPY-27 occupancy at the ectopic *rex-32* site on chromosome I was equivalent to DPY-27 occupancy at the *rex-32* site on X (*Figure 5B*).

Despite strong DCC binding to *rex* sites closely linked to the autosomal reporters, expression of seven of eight reporters was not significantly reduced compared to expression of the same reporters integrated at the same autosomal sites without a *rex* site (*Figure 5A*). Furthermore, in *sdc-2(RNAi)* XX animals with very low levels DCC binding, expression of six of eight autosomal reporters with closely linked *rex* sites was not elevated compared to expression of the same *rex*-linked autosomal reporters in control XX animals with high levels of DCC binding. These results indicate that strong DCC binding adjacent to a gene is generally not sufficient to regulate its expression. Furthermore, they are consistent with results showing that close proximity of DCC binding to either an endogenous X-linked gene or an engineered X-linked transgene was not necessary for the dosage compensation of the gene. Our results strongly support a model in which DCC binding causes global changes to the X chromosome, likely by remodeling X topology (*Crane et al., 2015*) to elicit chromosome-wide gene repression.

## Transgenes on X are expressed at half the level as transgenes on autosomes in XX animals, contrary to a chromosome-wide mechanism to upregulate X expression

Having characterized a set of X-linked and autosomal transgenes thoroughly, we could use them to assess whether a chromosome-wide mechanism of upregulation functions in *C. elegans* to balance expression between X chromosomes and autosomes, consistent with Ohno's hypothesis. If an X-linked transgene is controlled by both a dosage compensation mechanism, which halves X expression in XX animals, and an upregulation mechanism, which doubles X expression in both sexes as Ohno hypothesized, the per-copy expression of the X transgene in XX animals will be similar to that of an autosomal transgene (*Figure 1D* and *Figure 6A*). However, if the X-linked transgene is down-regulated by the dosage compensation machinery in XX animals but is not controlled by an upregulation mechanism that operates in both sexes, the transgene will be expressed at half the level of an autosomal transgene (*Figure 1C* and *Figure 6A*). Lastly, if the X chromosome is controlled by an Ohno-like upregulation mechanism, an X-linked transgene in a dosage-compensation-defective XX mutant or an X transgene that escapes dosage compensation in wild-type XX animals will be expressed at twice the level of an autosomal transgene. Without X-chromosome upregulation, these X and autosomal transgenes will be expressed at a similar level (*Figure 6A*).

We found that the average total expression of all X-linked transgenes in XX animals (2 copies) was 56% of the average total expression of all transgenes on autosomes (2 copies) (p<0.0001, 95% CI of the mean between 0.497 and 0.620) (*Figure 6A*, right panel). Thus, the relative level of transgene expression on X compared to autosomes differs significantly from the ratio of 1 predicted by an Ohno-like model of X chromosome-wide upregulation (One Sample t-test, p<0.001) and is not significantly different from the ratio of 0.5 predicted by the lack of a general upregulation

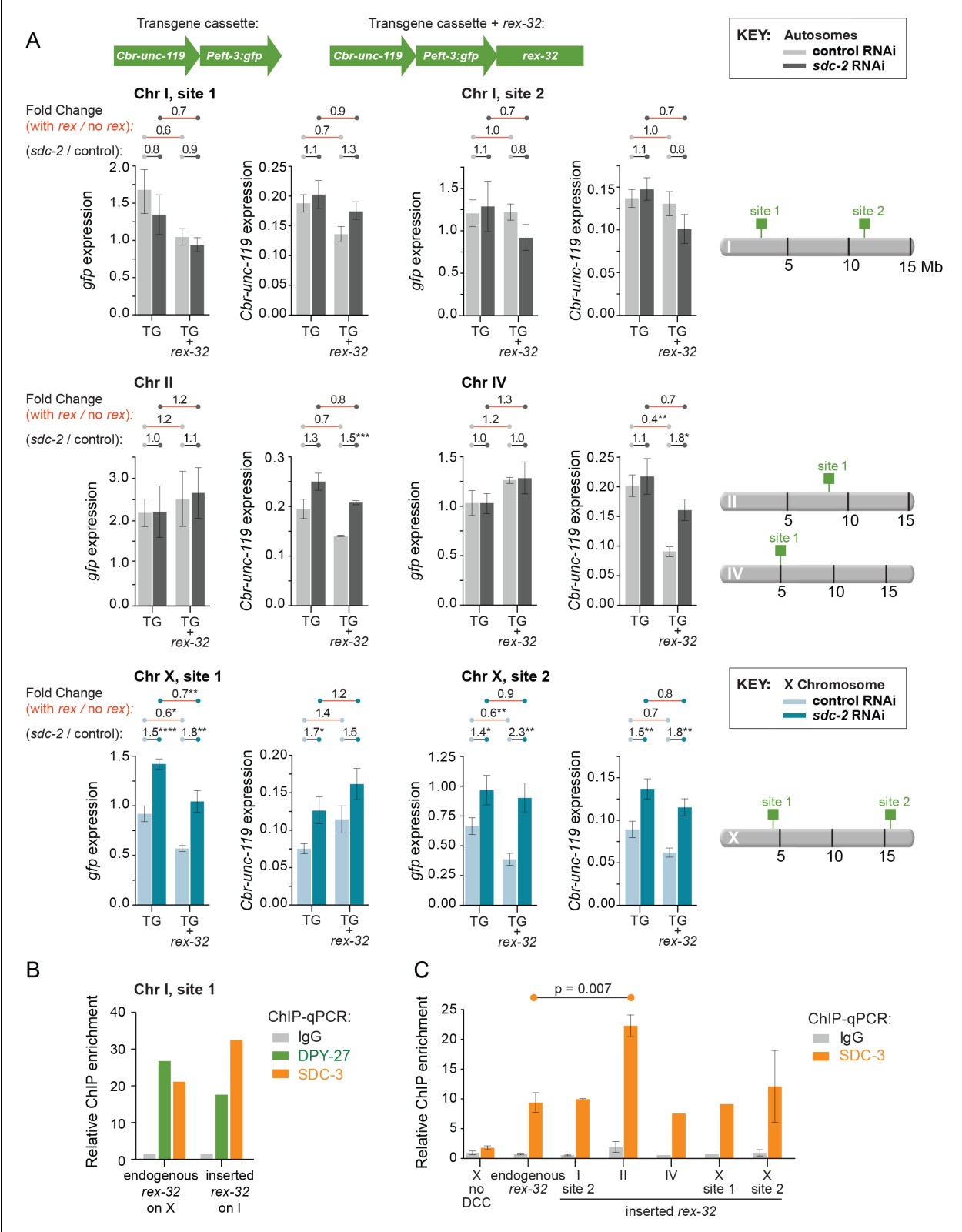

**Figure 5.** Comparison of transgene expression on X and autosomes with and without a co-inserted strong *rex* site. (**A**) Each graph depicts expression levels of either *Cbr-unc-119* or *Peft-3:gfp*, integrated at the same site on either X or an autosome with (TG + *rex-32*) or without (TG) a co-inserted copy of the strong DCC binding site *rex-32*, in both control RNAi XX (light) and *sdc-2(RNAi)* (dark) XX animals. The specific insertion site is indicated above the graph and corresponds to the schematic on the right. Expression of autosomal transgenes is shown in light and dark gray, and X-linked transgenes

*Figure 5 continued on next page*

*Figure 5 continued*

in light and dark blue. Numbers above the graphs show the fold change in gene expression (red lines) between transgenes with and without the co-inserted *rex* site in either control RNAi or *sdc-2(RNAi)* animals. Also shown is the fold change in expression (gray lines) of transgenes in control RNAi vs. *sdc-2(RNAi)* animals, either with or without *rex-32*. The number of asterisks indicates the p-value: p≤0.05, one asterisk; p≤0.01, two asterisks; p≤0.001, three asterisks; p≤0.0001, four asterisks. Proximity to a co-inserted *rex* site does not increase gene expression in *sdc-2(RNAi)* vs. control RNAi animals, nor does it generally decrease expression significantly on autosomes in control animals. Error bars show the standard error of the mean for at least three biological replicates. (**B**) Binding of DPY-27 (green), SDC-3 (orange), and IgG (gray) at Chr I, site 1 was assayed by ChIP-qPCR at the co-inserted copy of *rex-32*. For this site, ChIP was conducted in an engineered strain lacking the endogenous copy of *rex-32*, and the graph depicts the enrichment of the two DCC components at the center of the co-inserted *rex-32*. Similar ChIP experiments were conducted for the endogenous *rex-32* site in a wild-type strain. Enrichment is expressed relative to an autosomal region that does not recruit the DCC and is normalized to input. (**C**) For the designated sites on X and autosomes, SDC-3 (orange) and IgG (gray) ChIP were quantified at the inserted copy of *rex-32* in strains carrying the endogenous wild-type copy of *rex-32* by using primers that recognize the unique junction between *rex-32* and *Peft-3::gfp*. For four ectopic *rex* sites, the level of SDC-3 binding was similar to its level at the endogenous *rex-32* site. The ectopic *rex-32* inserted on Chr II bound significantly more SDC-3 than the endogenous copy on X. Error bars show the standard error of the mean for at least two biological replicates.

mechanism (One Sample t-test, p=0.06). Moreover, when we analyzed the data by the category of reporter gene (*Pdpy-30:gfp:H2B*, *Peft-3:tdT:H2B*, *Peft-3:gfp*, *Peft-4-gfp*, or *Cbr-unc-119*) and compared the average expression level for each reporter at all sites on X and on autosomes, we found that the average expression was also reduced for reporters on X compared to autosomes (**Figure 6B**), like the combined expression data for all reporters. Depending on the reporter, X chromosome transgenes were expressed on average between 44% and 68% of their autosomal counterparts (**Figure 6B**), again arguing against a chromosome-wide mechanism to increase gene expression on X chromosomes. For six of eight reporters tested, the reduction in gene expression on X vs. autosomes was statistically significant. The two remaining reporters *Peft-3:gfp* and *Peft-4: gfp* had reduced expression when integrated on X vs. autosomes, but the reduction was not statistically significant (fold change = 0.52, p=0.14; fold change = 0.68, p=0.13), likely due to the small number of these reporters on X. Together, these data argue against a chromosome-wide mechanism of X upregulation that increases gene expression on the two hermaphrodite X chromosomes to balance expression with that of autosomes.

In a separate analysis of Ohno's hypothesis, if a chromosome-wide mechanism were to elevate expression of genes on X, a transgene on the single male X chromosome would be expressed at twice the level as a transgene on one of a pair of homologous autosomes. We found to the contrary that expression of a single transgene on the male X was not different from expression of a single transgene on one of the autosomal homologs (**Figure 6C**). Thus, while our results demonstrate a DCC-mediated, chromosome-wide mechanism to equalize X gene expression between the sexes by reducing expression of endogenous and ectopic genes on hermaphrodite X chromosomes, our results argue against an analogous chromosome-wide mechanism that increases transcription of X chromosomes in both sexes to balance expression between X chromosomes and autosomes.

We were able to reach the robust conclusion that expression of transgenes on X is significantly lower than expression of transgenes on autosomes because our reporters had only minimal variability in expression when integrated at different sites along the X chromosome or autosomes. As examples, despite the diversity of insertion sites for the 14 *Cbr-unc-119* transgenes integrated across X or the 18 *Cbr-unc-119* transgenes integrated across autosomes, the variation in expression was low as indicated by the absolute variation in expression and the coefficient of variation (**Figure 3—figure supplement 1B**).

## Evidence against a speculative model of X-chromosome dosage compensation reliant on *rex*-dependent nuclear positioning of X

Our analysis of X-chromosome regulation enabled us to evaluate an attractive but speculative model of X-chromosome dosage compensation, which posits that repression of X-linked gene expression in XX animals by the DCC is merely the result of escaping from a chromosome-wide mechanism that upregulates X expression (*Sharma et al., 2014*; *Sharma and Meister, 2015*). In particular, these authors proposed that *rex* sites target X to the nuclear periphery in males to increase chromosome-

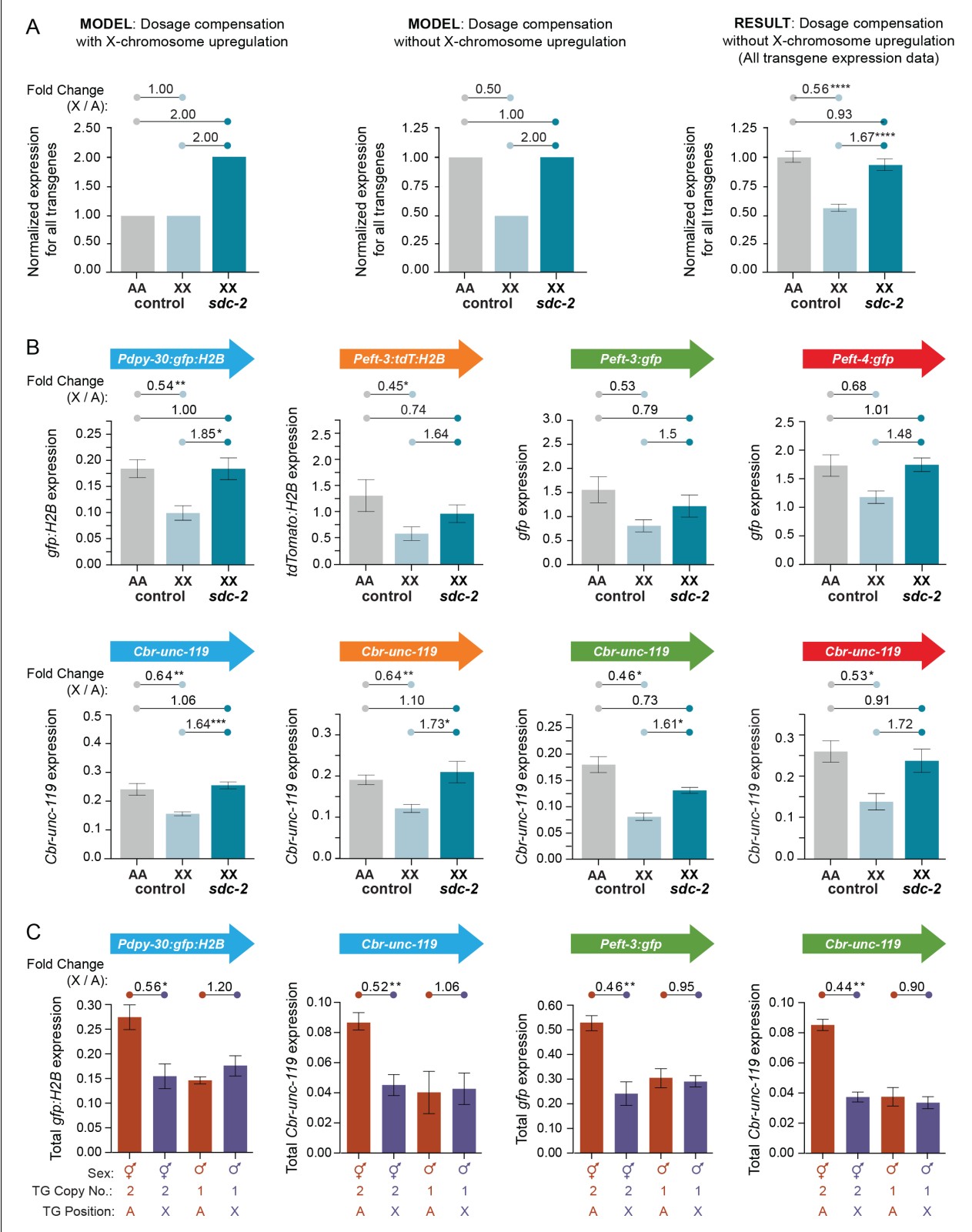

**Figure 6.** Transgenes on X are expressed at half the level of transgenes on autosomes. (**A**) Predicted vs. observed transgene expression levels for dosage compensated transgenes on X relative to transgenes on autosomes for two models of X-chromosome regulation. (Left) Under an X-chromosome-wide model of upregulation, dosage-compensated transgenes on hermaphrodite X chromosomes (2 copies) are predicted to have similar average total expression levels as transgenes on autosomes (2 copies), despite the hermaphrodite-specific repression by the DCC. Moreover,

*Figure 6 continued on next page*

*Figure 6 continued*

because the DCC reduces gene expression on X by about half, *sdc-2(RNAi)* animals would be predicted to have two-fold more transgene expression on X relative to autosomes, if X-chromosome upregulation occurred. (Middle) Under a model of no X-chromosome-wide upregulation, dosage compensated transgenes would be expressed at half the level of transgenes on autosomes due to repression by the DCC. In DCC-defective XX animals, transgene expression on X (2 copies) would increase to the level of transgene expression on autosomes (2 copies). (Right) The results of comparing the average expression level of all transgenes on X and autosomes argue against a chromosome-wide model of X-chromosome upregulation. For each reporter, data were normalized to the average autosomal expression level and then combined. Numbers above the graph show the fold change in expression between transgenes on X and on autosomes in control RNAi animals or between transgenes on X in *sdc-2(RNAi)* animals and transgenes on autosomes in control RNAi animals. The normalized expression level of all transgenes on the X chromosome (light blue) is only 56% of the normalized expression level of all transgenes on autosomes (gray). Expression of transgenes on X is increased to 93% of transgene expression on autosomes in animals treated with RNAi against *sdc-2* (dark blue). $p \leq 0.0001$, four asterisks. Error bars show the standard error of the mean. (B) Comparison of averaged mRNA expression for all reporter transgenes of each type located at all sites on X or autosomes as quantified in (A), except that expression levels were not normalized to the average autosome expression level. For example, in the first panel, averaged expression from four *Pdpy-30:gfp:H2B* reporter transgenes inserted on autosomes is compared with averaged expression from four *Pdpy-30:gfp:H2B* reporter transgenes inserted on X. Expression levels of the *Cbr-unc-119* reporter included with each transgene cassette are shown below the expression levels of the fluorescent partner transgenes. The number of asterisks indicates the p-value: $p \leq 0.05$, one asterisk; $p \leq 0.01$, two asterisks (Student's t-test). Error bars show the standard error of the mean. (C) Comparison between XX and XO L1/L2 larvae of total expression levels for transgenes inserted on X vs. autosomes. For the transgene cassette *Pdpy-30:gfp:H2B* and *Cbr-unc-119*, the autosomal cassette was on chromosome IV at site 1, and the X cassette was at site 3. For the transgene cassette *Peft-3:gfp* and *Cbr-unc-119*, the autosomal cassette was on chromosome I at site 1 and the X cassette at site 1. Shown below each bar are the sex of the animals in which gene expression was quantified, the copy number of the reporter transgene, and the position of the quantified reporter transgene (either on X or autosomes). For all transgenes, we compared the total level of expression from two reporter copies in XX animals and one reporter copy in XO animals, regardless of whether the reporters were on X or an autosome. The fold change in gene expression between reporters on X and autosomes is given above the graphs. The number of asterisks indicates the p-value: $p \leq 0.05$, one asterisk; $p \leq 0.01$, two asterisks (Student's t-test). Error bars show the standard error of the mean for at least three biological replicates. Expression of two copies on X was about half the expression of two copies on autosomes. Similarly, expression of the single copy on males was not different from the single copy on autosomes, meaning that expression of one copy on the male X would be half the expression of two copies on male autosomes. Results in (B, C) argue against an Ohno-like mechanism of X-chromosome upregulation in which chromosome-wide transcription of X is increased in expression. The results do not exclude the possibility that diverse gene-specific mechanisms might have arisen to elevate expression of individual X-linked genes with reduced dose.

wide gene expression, while DCC binding to *rex* sites in hermaphrodites blocks the peripheral localization, relocating X to the interior and hence reducing X gene expression.

The minimal evidence that led to this model included the following: (1) low-resolution DamID studies suggesting that X associates with nuclear pore proteins more frequently in males than in hermaphrodites, and FISH experiments suggesting that X associates with the nuclear periphery more frequently in XO than XX embryos; (2) FISH experiments suggesting that ectopic insertion of a truncated *rex* site at one autosomal location targets the locus to the nuclear periphery more frequently in XO and DCC-defective XX animals than in wild-type XX animals; (3) FISH experiments suggesting that seven X-linked *rex* sites had enriched association with the nuclear periphery in males vs. random nuclear positioning. However, only 3 of the *rex* sites (at the center of X: *rex-33*, *rex-28*, and *rex-8*) showed a significant increase in peripheral localization in males versus hermaphrodites.

No gene expression studies tested the validity of the model, and no DCC binding studies tested whether the truncated *rex* site integrated into the autosome recruited the DCC in XX embryos. The truncated autosomal *rex* site lacked a full-length DNA motif important for robust DCC binding, making the site unlikely to be a strong DCC recruiter in single copy (*McDonel et al., 2006*; *Jans et al., 2009*). In fact, the truncated site only partially recruited the DCC in vivo when present in multiple copies (*McDonel et al., 2006*).

Because our X-linked transgenes are fully responsive to the dosage compensation process, they are valid tools for assessing this untested model. Regulation of transgene expression should fulfill expectations of the model if the model is correct. Instead, results from four different experimental approaches, including transgene expression and chromosome localization studies, are inconsistent with expectations of this model. First, the nuclear positioning model predicts that in males the transgenes on X should have elevated expression relative to transgenes on autosomes due to a *rex*-dependent association of X with the nuclear periphery. However, we found that in males, expression of single-copy transgenes on the sole X chromosome was not different from expression of single-copy transgenes inserted on only one of two homologous autosomes (*Figure 6C*), contrary to the

nuclear positioning model of dosage compensation (*Sharma et al., 2014*; *Sharma and Meister, 2015*).

Second, the nuclear positioning model predicts that in XX animals defective in DCC binding, expression of a transgene co-inserted with a closely linked *rex* site should be higher than expression of the same transgene inserted at the same locus without a *rex* site. To test this prediction, we analyzed expression of the reporters *Cbr-unc-119* and *Peft-3:gfp* that were co-inserted with and without *rex-32* at four sites on autosomes and two sites on X. DCC binding to the ectopic *rex* sites was strong in the transgenic XX animals (*Figure 5B,C*).

We found that in XX animals, depletion of the DCC did not significantly increase the expression of any of the 12 *rex*-linked reporter transgenes on X chromosomes or autosomes compared to their expression at the same locations without a *rex* site (*Figure 5A*). These findings challenge the proposal (*Sharma et al., 2014*) that *rex* sites enhance expression of a nearby gene in the absence of DCC binding. Furthermore, loss of DCC binding in XX animals did not significantly elevate expression for six of eight autosomal reporters with an adjacent *rex* site (*Figure 5A*). Thus, DCC binding adjacent to a gene is not sufficient to induce repression of the gene, unlike the expectation from the nuclear positioning model.

Third, using the same approach as (*Sharma et al., 2014*) but contrary to their single result, we found that *rex* sites in autosomes, validated for DCC binding in wild-type XX animals, did not preferentially recruit flanking autosomal DNA to the nuclear periphery under conditions in which DCC binding was prohibited (*Figure 7 A–C*). We checked *rex* localization in young embryos of the age used by (*Sharma et al., 2014*) and also older embryos to be certain that embryo age did not affect our conclusion. Specifically, the autosomal regions of chromosome I and IV that flank the integrated ectopic *rex-32* site did not localize more frequently to the nuclear periphery (zone 1 vs. zones 2 and 3) in either younger (50–140 cells) or older (>200 cells) XO embryos (both lacking DCC binding at *rex-32*) compared to age-matched XX embryos (both proficient in DCC binding at *rex-32*) (*Figure 7 A–C*). In addition, autosomal loci adjacent to a *rex* insertion were not localized more frequently to the nuclear periphery in younger or older XO embryos than the same autosomal loci in age-matched XO embryos lacking the *rex* insertion. For comparisons involving younger embryos, p>0.5; for those involving older embryos, p>0.1 (chi-square tests). Thus, an ectopic *rex* site lacking DCC binding is insufficient to relocate flanking autosomal DNA to the nuclear periphery.

Fourth, all five endogenous *rex* sites tested on X, including three in the center (*rex-33, rex-47* and *rex-8*) and two toward one end (*rex-32* and *rex-23*), were not preferentially recruited to the nuclear periphery in XO embryos in either the 50-140-cell stage or the > 200-cell stage compared to age-matched XX embryos (p>0.1, chi-square test) (*Figure 7D* and *Figure 7—figure supplement 1 A,B*). Indeed, for *rex-23* the opposite was found: older XX embryos showed more peripheral localization than older XO embryos (p<0.001, chi-square test), but both sexes showed less peripheral localization than expected by random chance (p<0.0001). For *rex-47*, younger embryos of both sexes had enrichment in the peripheral-most zone (p=0.008), but peripheral enrichment in XO embryos was not greater than in XX. These results argue against a *rex*-dependent nuclear positioning model of dosage compensation. Furthermore, the inconsistency in *rex* nuclear localization across X in the *Sharma et al. (2014)* data set is difficult to reconcile with a spatial positioning model that is proposed to control gene expression across the entire X (see Discussion). We conclude that while we have not ruled out the interesting possibility that X chromosome nuclear positioning might play a role in *C. elegans* dosage compensation, we have provided compelling evidence that the explicit model of (*Sharma et al., 2014*) cannot account for the mechanism of *C. elegans* dosage compensation.

## Discussion

Changes in chromosome number have the potential to disrupt the balance of gene expression and thereby reduce organismal fitness. The evolution of sex chromosomes from autosomes provides an opportunity to dissect gene regulatory mechanisms that enable organisms to tolerate widespread imbalances in gene dose. In XX/XY and XX/XO species, X chromosomes retain many genes present on their autosomal ancestors, but Y chromosomes do not, leaving diploid males (XY or XO) with only a single copy of numerous genes. Without compensating mechanisms, genes on the male X would be expressed at half the total level as genes on the ancestral autosomes and on the two

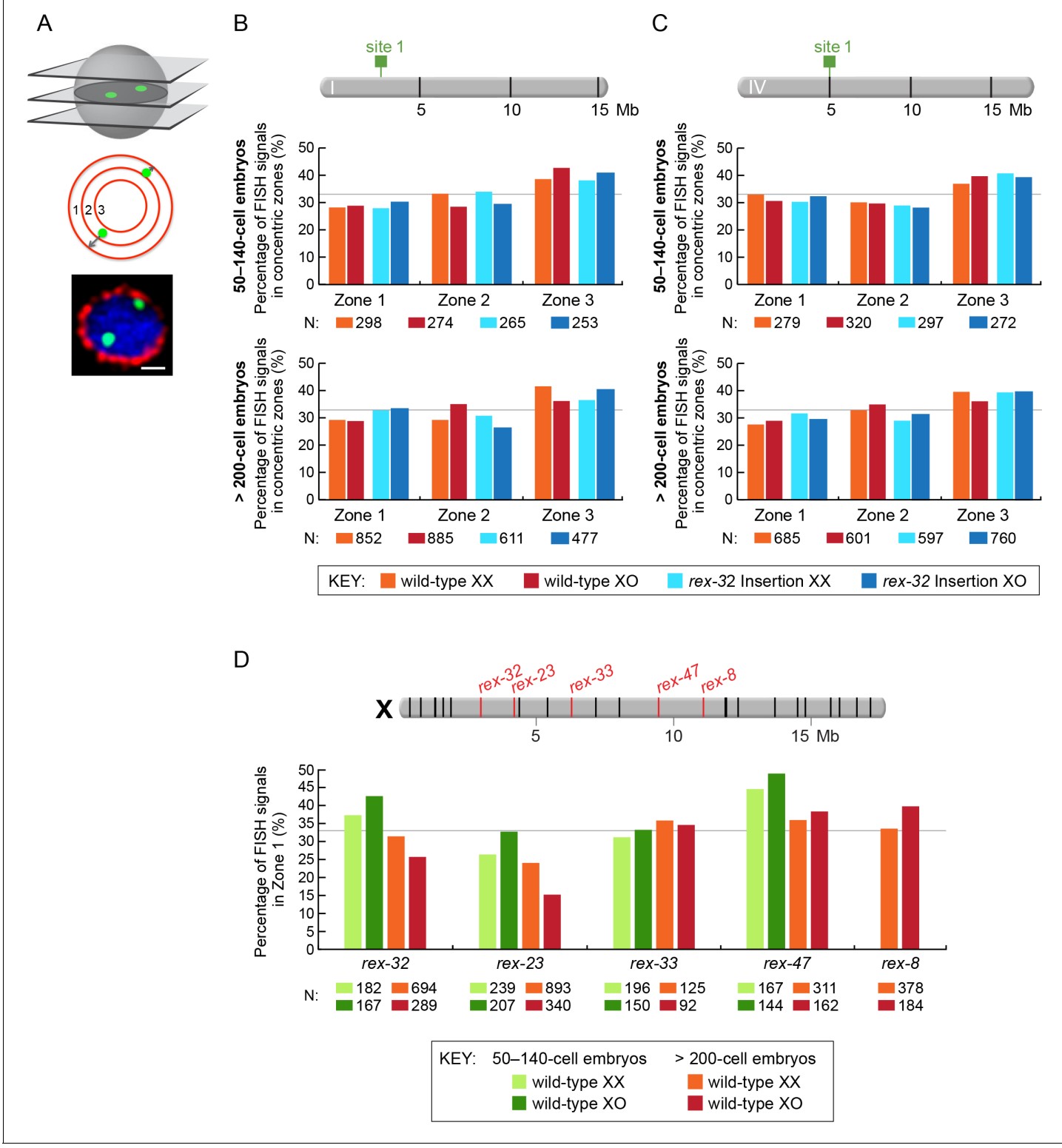

**Figure 7.** Proximity to an endogenous or ectopic *rex* sites does not determine the positions of X or autosomal loci relative to the nuclear periphery. (A) Illustration of approaches for quantifying the radial positions of FISH signals, the same approach as used by *Sharma and Meister (2014)*. (Top) A stack of confocal images determines the location of FISH signals in 3D. The nucleus is divided into three concentric zones with equal area. A random distribution would result in 33% of the FISH signals in each zone, as marked by the gray line in (B–D). (Middle), For each FISH signal, the ratio between its distance to the nuclear periphery and the nuclear radius determines the zone in which the site resides. (Bottom), Representative confocal image showing position of FISH signals in a nucleus. Blue, DAPI; Red, Nuclear Pore Complex; Green, *rex* FISH; Scale bar, 1 μm. (B, C) Histograms show the

*Figure 7 continued on next page*

*Figure 7 continued*

fraction of autosomal FISH signals in each of three zones in XX or XO embryos of two age groups (50-140-cell stage or > 200-cell stage) from wild-type strains or ectopic *rex*-insertion strains. Genomic locations of insertion sites (Chr 1, site 1 at position 2,85,041 Mb or Chr IV, site 1 at position 5,014,698 Mb) for transgenes with and without ectopic *rex* sites are shown on chromosome maps above the histograms. For both autosomal loci examined, nuclear positioning was not statistically different between (1) XX and XO embryos with an ectopic *rex* insertion; (2) between XO embryos with or without a *rex* insertion; (3) between XO embryos with a *rex* insertion and XX embryos without a *rex* insertion; or (4) between XX and XO embryos without a *rex* insertion, regardless of embryo age (p>0.5 for younger embryos and p>0.1 for older embryos, chi-square test). N is the total number of autosomal or *rex* FISH signals quantified for XX or XO embryos in the two age groups of each genotype. FISH probes were 30 kb. (D) Histograms show the fraction of endogenous *rex* FISH signals (*rex-32, rex-23, rex-33, rex-47,* or *rex-8*) at the nuclear periphery (zone 1) of wild-type XX or XO embryos of two age groups (50-140-cell stage or > 200-cell stage). Genomic locations of the 25 highest DCC-occupied *rex* sites on X are shown above the histograms (red, *rex* sites examined; black, all others). The positioning of *rex* sites at the nuclear periphery (zone 1) was not greater in XO versus XX embryos of either age group (chi-square test, see text).

The following figure supplements are available for figure 7:

**Figure supplement 1.** Radial position of *rex* sites on X relative to the nuclear periphery is not different between XX and XO embryos.

**Figure supplement 2.** Examples of XO embryos hybridized with FISH probes to assess nuclear positioning of ectopic *rex* sites integrated onto autosomes and endogenous *rex* sites on X.

**Figure supplement 3.** Examples of XO and XX embryos hybridized with FISH probes to assess nuclear positioning of *rex-33* on X.

contemporary female X chromosomes. Here we analyzed expression of single copy transgenes integrated on X chromosomes and autosomes to elucidate mechanisms that permit *C. elegans* to tolerate sex-chromosome-wide imbalances in gene dose.

Our analysis showed that the dosage compensation process, which equalizes X expression between the sexes by reducing expression of endogenous X-linked genes in hermaphrodites, acts on a chromosome-wide basis to reduce expression of all X-linked transgenes in XX animals, regardless of their source of promoter, position on X, or proximity to a dosage compensation complex (DCC) binding site. Not only is close proximity to a DCC binding site unnecessary to reduce X-linked transgene expression, it is also insufficient to repress expression of autosomal transgenes co-integrated with a DCC binding site. This finding reinforces the model that the *cis*-acting DCC regulatory sites act cumulatively to create an environment on X that is broadly permissive for repression of endogenous and engineered genes.

While reducing X expression in hermaphrodites balances gene expression between the sexes, it potentially leaves both sexes with inappropriately low expression of X chromosomes vs. autosomes. Our transgene expression studies enabled us to test whether a chromosome-wide transcription mechanism might elevate X expression in both sexes to balance gene expression between X chromosomes and autosomes, thereby facilitating X-chromosome evolution and fulfilling Ohno's hypothesis. We found that transgenes on hermaphrodite X chromosomes were expressed at half the level of their autosomal counterparts, contrary to a chromosome-wide mechanism to balance gene expression across the genome. Lastly, we found compelling evidence against an attractive but speculative model of X-chromosome dosage compensation that is contingent upon a non-sex-specific mechanism to elevate X expression chromosome-wide and a hermaphrodite-specific, DCC-dependent mechanism to inhibit X upregulation by preventing localization of X to the nuclear periphery.

## Chromosome-wide action of the dosage compensation complex

Whether DCC binding nearby a gene is necessary or sufficient for dosage compensation of the gene has been debated, leaving open the question of whether the dosage compensation mechanism acts on a gene-by gene basis or primarily through a chromosome-wide process that changes fundamental properties of X. Prior genome-wide studies supported a chromosome-wide mechanism by demonstrating no correlation between DCC binding in or near an endogenous gene and the dosage compensation status of the gene (*Jans et al., 2009*; *Kruesi et al., 2013*). Hence, DCC binding near a gene could not be the sole determinant of dosage compensation. An alternate interpretation by others posited that DCC binding near a gene is essential for its repression, but elevated expression

in dosage-compensation mutants of genes lacking a DCC binding site was an indirect consequence of the mis-regulation of other genes on X, instead of the attenuation of a chromosome-wide mechanism (*Strome et al., 2014*).

Our experiments provide strong evidence against this alternative interpretation. First, dosage compensation of transgenes integrated on X did not require local DCC binding or even close proximity to a strong DCC recruitment site. Dosage compensation was judged by two criteria: the increased expression of transgenes in dosage-compensation-defective XX mutants vs. wild-type XX animals, and the statistically indistinguishable levels of overall transgene expression between wild-type homozygous transgenic XX and hemizygous transgenic XO animals. The former criterion averted any potential differences due to sex-specific gene expression, and the latter criterion averted any complications that might arise from the disruption of dosage compensation and consequent elevation of X expression. Second, only transgenes on X were responsive to the dosage compensation process. Indirect effects caused by the disruption of dosage compensation would be predicted to affect transgenes on autosomes as well. Third, close proximity of strong DCC recruitment sites to transgenes on autosomes did not elicit gene repression. That is, in wild-type XX embryos, expression of transgenes on autosomes was not lower in the presence of nearby *rex* sites than in their absence, and transgenes co-inserted with *rex* sites were not generally increased in expression in dosage-compensation-defective mutants, further demonstrating that local DCC binding is not sufficient to establish dosage compensation.

The promoter of an X-linked gene also does not dictate the dosage compensation status. Although the *eft-4* gene is dosage compensated at its endogenous location on X, and X-linked transgenes driven by the *eft-4* promoter are dosage compensated, *eft-4*-driven transgenes on autosomes are not. Thus, the promoter does not transmit responsiveness to the dosage compensation process.

Our data support the model that *rex* sites act in concert and over long distance to establish an X-chromosome environment that promotes reduction of gene expression across the entire chromosome, even for endogenous and engineered genes that lack DCC binding sites. Our results are in strong agreement with recent studies demonstrating that DCC binding at *rex* sites controls the topology of the entire X chromosome (*Crane et al., 2015*).

## How do some endogenous X-linked genes escape dosage compensation?

All engineered transgenes on X were responsive to dosage compensation yet some endogenous X-linked genes escape dosage compensation in *C. elegans*. These results suggest a model in which endogenous X-linked genes that escape regulation may have special features that insulate them from repression by the dosage compensation machinery. These features likely operate locally because genes that escape dosage compensation are in close proximity to and interspersed with genes that undergo dosage compensation (*Jans et al., 2009*; *Kruesi et al., 2013*; *Crane et al., 2015*). As in *C. elegans*, most X-chromosome transgenes in both *Drosophila melanogaster* and mice are subject to XX/XY dosage compensation (*Scholnick et al., 1983*; *Spradling and Rubin, 1983*; *Krumlauf et al., 1986*; *Dandolo et al., 1993*; *Tan et al., 1993*; *Yang et al., 2012*), but in both species, some endogenous genes on X escape. In support of an insulation model, the territory surrounding the mouse gene Kdm5c has two separable regulatory domains that influence X inactivation in opposite ways: one causes Kdm5c to escape X inactivation at its endogenous site on X and at ectopic sites on X, and a second region prevents X-linked genes nearby Kdm5c from escaping X inactivation (*Horvath et al., 2013*). Whether similar insulators exist in *C. elegans* is yet to be determined.

## Evidence against an X-chromosome-wide mechanism of transcriptional regulation to balance gene expression between X chromosomes and autosomes

Upregulation of X-chromosome gene expression was proposed to be an essential step in the evolution of sex chromosomes from a pair of ancestral autosomes (*Ohno, 1967*). However, compelling evidence has neither validated nor refuted this hypothesis for most species, with the plausible exception of placental and marsupial mammals, which had different outcomes (*Julien et al., 2012*). Our analysis in *C. elegans* overcame limitations in prior studies and provided strong evidence against a

chromosome-wide mechanism that increases transcription on X to balance gene expression between X chromosomes and autosomes during sex-chromosome evolution. These results rule out one plausible molecular mechanism by which Ohno's hypothesis might work, suggesting that if upregulation does occur, it operates through multiple, diverse gene-specific mechanisms, as discussed later.

In hermaphrodites, the X-linked dosage-compensated transgenes (2 copies) were expressed at 56% of their autosomal counterparts (2 copies), a value significantly different from the X-to-A expression ratio of 1 predicted by a model of X-chromosome-wide upregulation. In males, expression of transgenes on X (1 copy) was indistinguishable from expression of transgenes on one homolog of an autosome pair (1 copy), in contrast to the doubled X expression predicted by X-chromosome-wide upregulation. Both observations are inconsistent with a chromosome-wide mechanism for elevating X-linked gene expression.

We propose that the discrepancy in conclusions about X-chromosome-wide upregulation between our current study and previous studies (*Deng et al., 2011*; *Kruesi et al., 2013*) results from an incorrect prior assumption that the average expression of all present-day autosomes serves as a reliable proxy for expression of proto-X chromosomes. That assumption was shown to be incorrect for placental mammals (*Julien et al., 2012*), and our observation that the average expression for each of the five autosomes varies by 1.9-fold undermines that assumption for *C. elegans*. Even under a conservative assumption that the proto-X chromosome (pX) would be expressed within the range of individual present-day autosomes, the predicted X/pX expression ratio would vary from 0.56 to 1.22. An X/pX expression ratio of 0.5 would refute Ohno's hypothesis, and a ratio of 1 would support it, making the expression level of present-day autosomes too dissimilar to estimate pX expression for a test of Ohno's hypothesis.

In a different test of Ohno's hypothesis, a recent *C. elegans* study compared the expression of 276 genes located on chromosome I in the nematode *Pristionchus pacificus* but on chromosome X in *Caenorhabditis* due to a chromosome translocation that occurred after the species' divergence (~ 300 MYA) (*Albritton et al., 2014*). The study found that in XO animals, the new genes on the *C. elegans* X chromosome were expressed, on average, at about half the level of their autosomal orthologs in *P. pacificus*, a result the study concluded is inconsistent with chromosome-wide upregulation of X. While this study bypassed the need to approximate the proto-X expression level, it did not take into account X-chromosome silencing during germ cell proliferation. Expression of X chromosomes and autosomes was measured in young adults, leaving open the possibility that the reduction in expression of 276 genes on the *C. elegans* X chromosome was due to X-chromosome-specific germline silencing, rather than lack of X-chromosome upregulation in somatic cells. Consistent with this interpretation, we found that expression of the 276 genes was higher in somatic cells of *C. elegans* germlineless XX L4 larvae than in the combination of somatic and germ cells found in fertile, wild-type L4 XX larvae (*Figure 1—figure supplement 1E*). Therefore, germline silencing, which causes an underestimate of somatic gene expression in both sexes (*Deng et al., 2011*), is a plausible cause for the apparent reduction in expression of genes naturally translocated to X during evolution compared to their expression when sited on autosomes. Furthermore, without knowledge of whether the autosomal *P. pacificus* genes are subject to germline silencing, the data also cannot be used to conclude that X upregulation occurs. Thus, the experimental design appears to have precluded a reliable assessment of gene expression needed to assess Ohno's hypothesis.

In other experiments bearing on nematode X-chromosome upregulation, the histone mark H4K16ac was used as a proxy for gene activity (*Wells et al., 2012*). In *Drosophila*, elevation of H4K16Ac is the hallmark of an upregulated male X chromosome. In the *C. elegans* studies, H4K16ac was found to be depleted on X chromosomes vs. autosomes of wild-type hermaphrodites and enriched on X vs. autosomes of dosage-compensation-defective hermaphrodites, suggesting an involvement of H4K16Ac in upregulation. However, males showed no X-chromosome enrichment of H4K16Ac, a result that precludes a role for H4K16Ac in X-chromosome upregulation. Thus, no prior evidence validates or convincingly refutes Ohno's hypothesis in *C. elegans*.

Our current work provides strong evidence against an X-chromosome-wide mechanism of transcriptional upregulation that would fulfill Ohno's hypothesis. The lack of global X upregulation in *C. elegans* is consistent with findings in placental mammals (*Julien et al., 2012*), suggesting that many organisms tolerated the evolution of sex chromosomes without a global compensation mechanism to correct for reduced X-chromosome gene expression. (See discussion below for alternative mechanisms of compensation).

## Analysis countering an X-chromosome dosage compensation mechanism based on *rex*-directed nuclear positioning of X

Existence of an X-chromosome-wide mechanism to upregulate gene expression was a key assumption in a model proposed to explain X-chromosome dosage compensation (*Sharma et al., 2014*; *Sharma and Meister, 2015*). In this nuclear positioning model, *rex* sites in males were proposed to promote interactions between X chromosomes and nuclear pore proteins at the nuclear periphery that would induce chromosome-wide upregulation of X expression. In hermaphrodites, DCC binding to *rex* sites was proposed to block interactions between X chromosomes and nuclear pore proteins, thereby reducing expression by preventing upregulation of both Xs.

Key predictions of this nuclear positioning model were not fulfilled by our studies. Transgenes on X that were shown to be fully responsive to the dosage compensation process were not regulated in a manner consistent with the nuclear positioning model of dosage compensation: for example, X-linked transgenes were not expressed at higher levels in males than transgenes on autosomes. Furthermore, ectopic *rex* sites on X chromosomes or autosomes were insufficient to alter gene expression of nearby transgenes or to relocate flanking DNA to the nuclear periphery. Moreover, all five *rex* sites tested on X failed to show enhanced peripheral localization in males compared to hermaphrodites. Included among them were three *rex* sites in the center of X, contrary to the positive results of Sharma and Meister, and two *rex* sites at the end of X, consistent with their negative results. Since the experimental approach was similar for both laboratories, reasons for differences in *rex* localization patterns are not apparent. More confounding for the nuclear positioning model is the finding by Sharma and Meister that differences in *rex* nuclear localization patterns between the sexes were not uniform across X, unlike the expectation for a robust mechanism of dosage compensation that acts chromosome wide.

## Possible mechanisms to compensate for gene dose reduction during sex-chromosome evolution

Without a compensating mechanism to upregulate X-chromosome-wide gene expression, how did organisms tolerate the reduction in dose of neo-X-linked genes during sex chromosome evolution caused by the loss of homologous genes on Y or the complete demise of Y? We consider several possibilities for *C. elegans.* First, hemizygosity of many X-linked genes might not have been deleterious during sex chromosome evolution. Precedent for this possibility comes from studies of the newly formed sex chromosomes (neo-X and neo-Y) of *Drosophila miranda.* In *D. miranda*, a fusion between the Y chromosome and an autosome initiated the formation of these new sex chromosomes about 1 MYA (*Bachtrog and Charlesworth, 2002*). Since the fusion, almost half of the genes on the neo-Y chromosome were lost or accumulated inactivating mutations (*Bachtrog et al., 2008*; *Zhou and Bachtrog, 2012*). The neo-X chromosome acquired dosage compensation by upregulating X-chromosome expression in males using the same dosage compensation machinery as in *D. melanogaster* (*Bone and Kuroda, 1996*; *Marín et al., 1996*; *Alekseyenko et al., 2013*). However, compensation in *D. miranda* is far less complete than that for *D. melanogaster* (*Zhou et al., 2013*). Many genes that were lost from the neo-Y chromosome are not yet upregulated on the *D. miranda* neo-X chromosome, suggesting that *D. miranda* can tolerate hemizygosity of many X-chromosome genes in the male XY sex.

A second mechanism to compensate for reduced X-chromosome gene dose might have been to alter the genetic content of X chromosomes during sex chromosome evolution to favor genes whose lowered dose in males would be tolerated. In *C. elegans*, gene content on X differs from that on autosomes: the X is significantly depleted of essential and haploinsufficient genes compared to autosomes (*Kamath et al., 2003*; *de Clare et al., 2011*; *Albritton et al., 2014*). Furthermore, a few genes on the *C. elegans* X chromosome maintain a functional paralog on an autosome (*Maciejowski et al., 2005*). Together, these features of X have been proposed to prevent problems caused by germline silencing of X, but changes in X-chromosome gene content may equally well have played a role during sex chromosome evolution to accommodate the reduced dose of X-linked genes. Relocating haploinsufficient or essential genes from X to autosomes may have relaxed the pressure to balance gene expression between X chromosomes and autosomes.

Third, instead of one chromosome-wide mechanism of gene regulation, several different gene-specific mechanisms might have compensated for the reduced dose of haploinsufficient genes on X.

Precedent for this model comes from studies of aneuploidy in yeast and dose-sensitive genes in placental mammals. A balance hypothesis, borne out in yeast, posits that gene dose is especially important for genes that encode subunits of protein complexes, because changes in gene dose can disrupt complex formation by altering subunit stoichiometry (*Papp et al., 2003*). In recent studies of aneuploid laboratory yeast strains, post-translational mechanisms, especially protein degradation, were found to attenuate the increase in protein levels caused by the increased dose of specific classes of genes, particularly those encoding subunits of multi-protein complexes (*Torres et al., 2010*; *Dephoure et al., 2014*). In wild yeast strains, chromosomal amplifications have been found, but a debate exists over whether an active mode of dosage compensation specifically reduces transcript levels of amplified genes (*Hose et al., 2015*; *Gasch et al., 2016*; *Torres et al., 2016*).

In placental mammals, which lack global X upregulation, reduced X-chromosome dose can be compensated in some cases by increasing expression of dose-sensitive genes on X, particularly those producing subunits of large protein complexes (*Pessia et al., 2012*), or by decreasing expression of autosomal genes that produce subunits of macromolecular complexes containing proteins encoded by X-linked genes (*Julien et al., 2012*). While the general mechanisms that govern local changes in mammalian gene expression are not well known, one source of gene regulation is the mammalian histone methyl transferase complex called MOF, which acetylates histone H4 on lysine 16 to upregulate expression of a small set of genes on X (*Deng et al., 2013*). In addition, for some X-linked genes, evidence also exists for enhanced mRNA stability or enhanced translational efficiency via increased ribosome density as possible mechanisms to compensate for reduced gene dose (*Deng et al., 2013*; *Faucillion and Larsson, 2015*).

Some combination of mechanisms similar to those used in yeast and mammals might also function in *C. elegans* to compensate for dose-sensitive genes on X. Indeed, if loss of genes from sex chromosomes were gradual, the genome would have had the opportunity to respond in a gene-by-gene fashion to compensate for the reduced X gene dose, thereby rendering a global mechanism of X-chromosome upregulation unnecessary.

## Materials and methods

### Strains

All *C. elegans* strains were derived from the Bristol N2 variant and were maintained as described in (*Brenner, 1974*). *Supplementary file 1* contains a complete list of strains used in this study.

### Oligos

*Supplementary file 2* contains a complete list of oligos used in this study.

### MosSCI and miniMos

Strains were constructed and transgene copy number was analyzed as in (*Frøkjær-Jensen et al., 2008*, *2014*). The hallmark of multi-copy transgenes and transgenes resulting from imprecise insertion events is the incorporation of DNA from the cloning vector backbone into the worm genome. Therefore to obtain strains with single-copy transgenes, we performed PCR with primers specific to the vector backbone to identify and eliminate strains that carried complex transgene insertion events.

### RNAi and isolation of L1 hermaphrodites

Each 50 ml LB culture was supplemented with ampicillin (10 µg/ml) and inoculated with Ahringer feeding library bacteria bearing an *sdc-2* plasmid or, as a control, a plasmid with no insert (*Kamath et al., 2001*). Cultures were grown at 37°C for 12–16 hr and concentrated 10-fold. RNAi plates (1 mM IPTG, 25 µg/ml carbenicillin) were inoculated with 200 µl concentrated bacteria and incubated at 25°C for 24 hr. Gravid hermaphrodites of the appropriate genotype were bleached, and 800 embryos were plated on each control and *sdc-2(RNAi)* plate and incubated at 20°C. After 4 days, RNAi plates were washed twice with 5 ml M9 (first for 3 min, then for 1 min) to remove all hatched worms but retain embryos. Plates were then returned to 20°C for three hours to permit embryos to hatch. The hatching synchronized L1s were collected by washing the plate with 5 ml M9. To remove any embryos that may have become dislodged from the plate during the final wash, L1s

were concentrated and contaminating embryos were removed by mouth pipetting while viewing the animals with a dissecting microscope. L1s were frozen on liquid nitrogen and stored at -80°C. For each transgene, data were collected from at least three independent biological replicates.

## Harvesting L1/L2 males and hermaphrodites

To isolate males bearing a transgene of interest, approximately 200 L4 hermaphrodites of genotype *him-8(e1489) IV; unc-58(e655dm) X* were mated for 2 days at 20°C with approximately 200 males carrying the appropriate transgene. Since *unc-58(e655dm)* causes a dominant paralyzed phenotype, and *him-8(e1489)* hermaphrodites produce nullo-X oocytes due to X-chromosome non-disjunction, only transgene-bearing XO male cross progeny that inherited a paternal X chromosome will be mobile due to the lack of the *unc-58* mutation. To collect these mobile males, embryos were bleached and spotted on the empty half of a plate that had OP50 bacteria cultured on the other half. After 18 hr, L1/L2 male cross-progeny were isolated by cutting the plate in half and collecting only the larvae that had crawled onto the bacteria. To collect stage-matched homozygous hermaphrodite controls, embryos were collected by bleaching non-mated gravid hermaphrodites and isolated as above. Larvae were frozen on liquid nitrogen and stored at -80°C.

## RNA isolation and cDNA synthesis

RNA isolation was conducted as described in (*Baugh et al., 2003*) and at www.mcb.harvard.edu/hunter with two modifications: 1 ml of TriZOL reagent (Life Technologies [Carlsbad, CA] 15596–026) was used instead of 300 µl, and the pellets were resuspended in 12 µl of nuclease-free water. cDNA was made from 6 µl RNA for the L1 hermaphrodites (10 µl RNA for the L1/L2 hermaphrodites and males) using the QuantiTect Kit (Qiagen [Hilden, Germany] 205313). A no-reverse-transcriptase control was generated using 2 µl RNA.

## Quantitative PCR

Gene expression was analyzed using SYBR green (BioRad [Hercules, CA] iQ SYBR Green Supermix 170–8886) on a BioRad CFX384 Real-Time System. Standard curves were generated from genomic DNA and expression levels were determined from the appropriate standard curve by CFX detection software. Plus and minus reverse-transcriptase reactions were diluted 1/6 for L1 hermaphrodites and 1/2 for L1/L2 hermaphrodites and males, and 2 µl diluted cDNA was used as template in each 10 µl qPCR reaction. Each cDNA sample was quantified in triplicate with normalization and query primer sets.

To select normalization genes that are stably expressed in control RNAi and *sdc-2(RNAi)* conditions, we required that normalization gene candidates meet two conditions. First, the gene must be located on an autosome. Second, the normalization genes must not be significantly different in *sdc-2(y93, RNAi)* embryos as assessed by RNA-seq, GRO-seq, or microarray expression analysis (*Jans et al., 2009*; *Kruesi et al., 2013*; *Crane et al., 2015*). For the 12 candidate normalization genes that met the first two criteria, we used the geNorm approach (*Vandesompele et al., 2002*) to narrow normalization genes to those that are the most stably expressed in L1s, as isolated above. This approach indicated that *cdc-42, H06O01.1*, and *Y38A10A.5* were the best normalization genes. *gfp, Cbr-unc-119*, and *tdTomato* expression levels were normalized to the geometric mean of three normalization primers.

To quantify changes in gene expression, we compared the average of at least three biological replicates of control or *sdc-2(RNAi)* animals. Error bars represent the standard error of the mean and p values were generated using a Student's t-test (Graphpad prism).

## RNA-seq

To quantify average transcript levels on X and autosomes we used RNA-seq data generated in (*Crane et al., 2015*) and similar protocols for analysis. Libraries were sequenced with Illumina's HiSeq 2000 platform. Reads were required to have passed the CASAVA 1.8 quality filtering to be considered further. To remove and trim reads containing the sequencing barcodes, we used cutadapt version 0.9.5 (https://cutadapt.readthedocs.org/) (*Martin, 2011*). Reads were aligned to the WS220 transcriptome using GSNAP version 2012-01-11 (*Wu and Nacu, 2010*). Uniquely mapping reads were assigned to genes using HTSeq version 0.5.4p3 using the union mode (*Anders et al.,*

*2015*). We used DESeq to calculate normalization factors and for significance testing (*Anders and Huber, 2010*). To calculate expression level of individual genes, the normalized expression values from DESeq were divided by gene length (kb). Gene expression boxplots were generated using either all genes with a normalized expression level greater than zero or genes within the top 90% of expressed genes.

Raw reads from RNA-seq experiments in L4 animals with and without germlines were downloaded from the NIH short read archive (*Deng et al., 2011*). For the N2 transcriptome, we used the files SRR023579.sra, SRR023580.sra, and SRR023581.sra. For the *glp-1(q224)* germlineless transcriptomes, we used the files SRR031122.sra and SRR031123.sra. Reads were processed using the approach above.

## ChIP

Wild-type N2 animals were grown on NG agar plates with HB101 bacteria. Mixed-stage embryos were harvested from gravid hermaphrodites, and cross-linked with 2% formaldehyde for 10 min. Cross-linked embryos were resuspended in 1 ml of FA buffer (150 mM NaCl, 50 mM HEPES-KOH (pH 7.6), 1 mM EDTA, 1% Triton X-100, 0.1% sodium deoxycholate, 5 mM DTT, protease inhibitor cocktail, 1 mM PMSF) for every 1 g of embryos. This mixture was frozen on liquid nitrogen, then ground under liquid nitrogen by mortar and pestle until few intact embryos were visible with a dissecting microscope. Chromatin was sheared by the Covaris S2 sonicator (20% duty factor, power level 8, 200 cycles per burst) for a total of 30 min processing time (60 s ON, 45 s OFF, 30 cycles).

To perform the ChIP reactions, extract containing approximately 40 μg of DNA was incubated in a microfuge tube with 6.6 μg of anti-DPY-27, anti-SDC-3 or random IgG antibodies overnight at 4°C. A 50 μl bed volume of protein A Sepharose beads was added to the ChIP for 4 hr. ChIPs were washed for 5 min at room temperature twice with FA buffer (150 mM NaCl), once with FA buffer (1 M NaCl), once with FA buffer (500 mM NaCl), once with TEL buffer (10 mM Tris-HCl (pH 8.0), 250 mM LiCl, 1% NP-40, 1% sodium deoxycholate, 1 mM EDTA), and twice with TE buffer (10 mM Tris pH8, 1 mM EDTA). Protein and DNA were eluted twice with 1% SDS, 250 mM NaCl, 1 mM EDTA at 65°C for 15 min. After reversing crosslinks overnight at 65°C, and treatment with proteinase K and RNAse A, DNA was isolated using the Qiagen PCR purification kit. Using quantitative PCR, DCC enrichment was calculated at a query locus relative to a non-DCC bound autosomal control and is plotted relative to input DNA.

## FISH

To assess the nuclear positioning of the ectopic *rex* insertions *oxSi239* and *oxSi246* on autosomes, FISH probes were made using the following fosmids (BioScience LifeSciences, Nottingham, United Kingdom) that are adjacent to or flanking the transgene insertion sites: fosmid WRM062dG04 for *oxSi239* at chromosome I: 2,813,818–2,846,292 Mb and fosmid WRM0633bA08 for *oxSi246* at chromosome IV: 5,005,044–5,0426,76 Mb. The probes were labeled with Alexa-594 using FISH Tag DNA Kit (Invitrogen).

To assess the nuclear positioning of endogenous *rex* sites in older embryos, FISH probes were made from PCR products corresponding to ~5 kb regions surrounding *rex-32, rex-23, rex-47* or *rex-8*. The primers used are as follows: *rex-23* F (gcccattcaacccattgtcc); *rex-23* R (gcactcgcatattc-caaaacg); *rex-32* F (cgcagctggccgttaaatg); *rex-32* R (cattgcaggtgcgttcacaac); *rex-47* F (ccgaaacacaa-caacaatgc); *rex-47* R (agactggcgaagaggaacaa); *rex-8* F (tgtgatgcaagccagagttgg); *rex-8* R (cattgagccgaatttccaaagg). For younger embryos, the FISH probes were made from 30 kb fosmid probes as follows: *rex-47*, WRM0631aB04; *rex-23*, WRM0626cG08; *rex-32*, WRM0638aF07. To assess the nuclear position of the endogenous *rex-33* site in both younger and older embryos, FISH probes were made to two overlapping fosmids (WRM0615aA09 and WRM063cD06) that were labeled with different fluorescent dyes. Quantification was performed only on FISH spots that had signals from both fosmids to unambiguously identify *bona fide* FISH spots.

*C. elegans* embryos were obtained for *Figure 7B–C* by dissecting three different strains of mated gravid adult hermaphrodites: wild-type (N2), TY5726, and EG6136. Older *C. elegans* embryos were obtained for *Figure 7D* and *Figure 7—figure supplement 1B* by dissecting gravid adult hermaphrodites from strain CB1489 *him-8(e1489).* Younger embryos were obtained by dissecting gravid adult wild-type hermaphrodites that had been mated with wild-type males.

FISH was performed on the embryos as described previously (*Crane et al., 2015*). Following FISH, immunostaining with rabbit DPY-27 antibody (rb699) (*Chuang et al., 1994*) and Alexa-Fluor-647 goat anti-rat antibody (Invitrogen) was performed to determine the sex of embryos for experiments involving ectopic *rex* sites on autosomes and older embryos for endogenous *rex* sites on X. Embryos were determined to be XX if they exhibited punctate DPY-27 staining on both X chromosomes and XO if they lacked DPY-27 staining. Sex was determined for all embryos examining *rex-33* localization and all younger embryos examining localization of endogenous *rex* sites on X by counting the number of FISH spots in the nuclei. Embryos were determined to be XX if the nuclei had two FISH spots and XO if nuclei had one FISH spot. The age of embryos was determined by counting the number of DAPI-stained nuclei using the FindPoints function in Priism software (*Chen et al., 1996*). Confocal image stacks with a 51.5 nm XY pixel size and an 83.9 nm Z-spacing were obtained on a Leica TCS SP8 microscope using a 63×, 1.4 NA objective lens. Our Z-spacing was smaller than that used in (*Sharma et al., 2014*), making our point picking more precise. Their XY pixel size was not available for comparison. Image deconvolution with a theoretical point spread function was performed using Huygens Professional Software (Scientific Volume Imaging, The Netherlands). FISH spots were identified using the FindPoints function. To unambiguously select *bona fide* FISH spots, we excluded nuclei that showed more than one or two spots for XO and XX embryos, respectively. The nuclear positioning of the FISH signals was measured using a previously described approach (*Meister et al., 2010*), but with the Pick Points function in Priism software. Spots at the top or bottom of nuclei were excluded.

## Acknowledgements

We thank the Vincent J Coates Genomics Sequencing Laboratory at UC Berkeley, supported by NIH S10 Instrumentation Grants S10RR029668 and S10RR027303, for Illumina sequencing; S Uzawa for advice about FISH; D Stalford for assistance with figures; A Wood for developing reporter plasmids; N Fuda for performing metagene analyses in *Figure 1—figure supplement 1D*; and K Brejc, T Cline, B Farboud, N Fuda, and T Lee for discussions and comments on the manuscript.

## Additional information

### Funding

| Funder | Grant reference number | Author |
| --- | --- | --- |
| National Institutes of Health | 1F32 GM100647 | Bayly S Wheeler |
| Howard Hughes Medical Institute | | Christian Frøkjær-Jensen Barbara J Meyer |
| National Institutes of Health | 1R01 GM095817 | Erik Jorgensen |
| National Institutes of Health | 1R01 GM030702 | Barbara J Meyer |

The funders had no role in study design, data collection and interpretation, or the decision to submit the work for publication.

### Author contributions

BSW, Conception and design, Acquisition of data, Analysis and interpretation of data, Drafting or revising the article; EA, Acquisition of data, Analysis and interpretation of data, Drafting or revising the article; CF-J, Conception and design, Acquisition of data; QB, Acquisition of data, Analysis and interpretation of data; EJ, BJM, Conception and design, Analysis and interpretation of data, Drafting or revising the article

### Author ORCIDs

Barbara J Meyer, http://orcid.org/0000-0002-6530-4588

# Additional files

## Supplementary files

• Supplementary file 1. Strains used in this study.

• Supplementary file 2. Oligos used in this study.

## Major datasets

The following previously published datasets were used:

| Author(s) | Year | Dataset title | Dataset URL | Database, license, and accessibility information |
|---|---|---|---|---|
| Deng X, Hiatt JB, Nguyen DK, Ercan S, Sturgill D, Hillier LW, Schlesinger F, Davis CA, Reinke VJ, Gingeras TR, Shendure J, Waterston RH, Oliver B, Lieb JD, Disteche CM | 2011 | SRR023579.sra | http://sra.dnanexus.com/runs/SRR023579 | Publicly available at the Sequence Read Archive + (accession no: GSE20136) |
| Deng X, Hiatt JB, Nguyen DK, Ercan S, Sturgill D, Hillier LW, Schlesinger F, Davis CA, Reinke VJ, Gingeras TR, Shendure J, Waterston RH, Oliver B, Lieb JD, Disteche CM | 2011 | Transcriptome Analysis of roundworm | http://www.ncbi.nlm.nih.gov/sra/?term=SRR023580 | Publicly available at the NCBI Short Read Archive (accession no: GSE20136) |
| Kreusi WS, Core LJ, Waters CT, Lis JT, Meyer BJ | 2013 | Condensin Controls Recruitment of RNA Polymerase II to Achieve X-Chromosome Dosage Compensation | http://www.ncbi.nlm.nih.gov/geo/query/acc.cgi?acc=GSE43087 | Publicly availale at the NCBI Gene Expression Omnibus (accession no: GSE43087) |
| Deng X, Hiatt JB, Nguyen DK, Ercan S, Sturgill D, Hillier LW, Schlesinger F, Davis CA, Reinke VJ, Gingeras TR, Shendure J, Waterston RH, Oliver B, Lieb JD, Disteche CM | 2011 | SRR031122.sra | http://sra.dnanexus.com/runs/SRR031122 | Publicly available at the Sequence Read Archive + (accession no: GSE20136) |
| Deng X, Hiatt JB, Nguyen DK, Ercan S, Sturgill D, Hillier LW, Schlesinger F, Davis CA, Reinke VJ, Gingeras TR, Shendure J, Waterston RH, Oliver B, Lieb JD, Disteche CM | 2011 | SRR031123.sra | http://sra.dnanexus.com/runs/SRR031123 | Publicly available at the Sequence Read Archive + (accession no: GSE20136) |
| Crane E, Bian Q, McCord RP, Lajoie BR, Wheeler BS, Ralston EJ, Uzawa S, Dekker J, Meyer BJ | 2015 | Condensin-Driven Remodeling of X-Chromosome Topology during Dosage Compensation | http://www.ncbi.nlm.nih.gov/geo/query/acc.cgi?acc=GSE59716 | Publicly available at the NCBI Gene Expression Omnibus (accession no: GSE59716) |

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
