## [Decision Letter]

Thank you for submitting your article "Chromosome-wide mechanisms to decouple gene expression from gene dose during sex-chromosome evolution" for consideration by *eLife*. Your article has been favorably evaluated by Kevin Struhl as the Senior editor and three reviewers, one of whom is a member of our Board of Reviewing Editors. The reviewers have opted to remain anonymous.

The reviewers have discussed the reviews with one another and the Reviewing Editor has drafted this decision to help you prepare a revised submission.

This manuscript addresses dosage compensation strategies between XX females and XO males in *C. elegans* using transgenes. Two types of imbalance occur due to the sex chromosomes in *C. elegans*: one between the sexes and the other between the single X copy in males and the autosomes present in two copies. Dosage compensation between the sexes is achieved by repression of each X chromosome in XX hermaphrodites by the well-studied DCC (dosage compensation complex) recruited to *rex* sites on the X chromosomes. Dosage compensation between autosomes and X chromosome is less well characterized. Using a large panel of single copy transgenes (two different reporters under three different gene promoters in various combinations located) on the X chromosome (28 transgenes) or on autosomes (36 transgenes), the authors investigate transcriptomes from XX and XO individuals for evidence of either type of dosage compensation. RNA Seq is used to assess the extent to which transgenes shows differences in expression when X-linked or autosomal, in XX versus XO individuals. L1 larvae taken within 3 hours of hatching were examined, as these do not yet have germ cells, thus avoiding the problem of X-chromosome silencing which would interfere with the accuracy of X/A expression measurements.

First, the authors demonstrate that the DCC can spread to all transgenes inserted on the X. This result is validated using homozygous DCC-deficient XX animals. Intriguingly the presence of a dosage complex binding site (DCC), whether a dox or a *rex* site, is not required for dosage compensation for tgs on the X chromosome and has no impact on expression of most tgs on autosomes.

Second, the authors examine the extent to which X-chromosome transgene expression is balanced between the X and autosomes (the Ohno hypothesis). The authors find that most X-linked reporter transgenes examined are expressed at half the level of autosomes. The authors propose that this goes against Ohno's hypothesis for X-chromosome-wide mechanism to balance gene expression between the X and autosomes.

Third, the authors examine a recently proposed model that X-chromosome dosage compensation is based on *rex*-dependent nuclear positioning of the X chromosome. Using their reporter transgenes, the authors find no evidence that localization at the nuclear periphery in XO males is linked to increased X-chromosome expression, versus decreased binding of DCC at *rex* sites and internal relocalisation in XX animals.

Overall, the manuscript is well written and communicates the experiments and results clearly. The main issue concerns the interpretation of some of the data particularly for the second and third sections. To summarise, the authors provide good evidence to support their first claim, i.e., that reduction of X-linked gene expression operates mechanistically on a chromosome-wide basis. The single copy transgene data is compelling and confirms previous genome wide studies that demonstrated that DCC sites near endogenous X-linked genes are neither necessary nor sufficient for dosage compensation between the sexes.

The authors' second claim that their evidence argues against Ohno's hypothesis of X upreglation needs some clarification and rewording to make it clear that the experiments address the specific molecular model of a chromosome-wide mechanism of upregulation, which like dosage compensation, can extend to transgenes. As the authors have used transgenes, one interpretation of their data is that whatever the mechanism of up-regulation of the X relative to autosomes, this does not spread to transgenes inserted on the X. In fact, as different strategies of up-regulation may apply to different X-linked genes, it is difficult to probe this using transgenes. Furthermore, results from transgenes cannot really be used to deduce whether endogenous X-linked genes are up-regulated during evolution as the authors propose. Ohno's original text and hypothesis does not state that X up-regulation occurred on a chromosome-wide basis. Ohno hypothesized an up-regulation of X-linked genes during millions of years of sex chromosome evolution prior to the evolution of repression of X-linked gene expression in females. Inserting a transgene on the autosomes or X chromosome and requiring an immediate up-regulation of the X-linked copy in essentially one generation cannot be a valid test of this hypothesis.

Finally, the third set of conclusions, concerning the role of nuclear positioning in X regulation is subject to the same caveats as above: the behavior of transgenes may not reflect endogenous X genes. Therefore, the authors should explain better why the use of transgenes is an appropriate means to test the model. The authors do look at three endogenous *rex* sites, however, and do not come to the same conclusions as a previous study by the Meister lab, although the reasons for this difference are not clear and require further investigation.

Thus, although this work provides novel insight about the mechanism of X-linked gene regulation in *C. elegans*, it cannot completely test Ohno's hypothesis of evolutionary X up-regulation. The manuscript (title, Abstract, text, and figures) will need to be revised accordingly and a number of specific points that should also be addressed, as listed below.

Specific points to be addressed:

1) Although the use of L1 larvae is clearly very elegant strategy to avoid the inclusion of germ cells which have a silent X, and to avoid potential variation in expression linked to distinct developmental lineage, they also introduce a potential caveat, which is that at such an early stage, X/A gene expression balance may not in fact be in place. The authors cannot rule out that at later stages there is global or local, X-linked up regulation to balance expression with autosomes. The authors should discuss this. One way to confirm this would be to use older animals from which the germ cells have been mechanically removed or genetically depleted.

2) The authors do not adequately explain their choice of genes and promoters in the inserted transgenes. How and why were the promoters for the GFP fusions chosen? Why is the promoter-driven GFP or *tdTomato* sometimes fused to histone H2B? Since all the conclusions of the manuscript are based on this transgene expression data, the authors should provide explicit justifications for each of the genes and promoters chosen. Also the authors should mention how tg copy number was verified to be single copy – how was this quantified?

3) The authors make the general conclusion that DCC binding adjacent to a transgene inserted on autosomes is not sufficient to regulate its expression. However Figure 5 shows that this is the case for 7/8 transgenes, which means that this statement is valid for 87% of transgenes tested, not necessarily all. The statement should be changed to "most" not "all".

4) For the DNA FISH data, it was unclear why the probe size used changed (Figure 7 legend), depending on the age of the embryos examined. Also, it would be desirable that the authors show examples of FISH data for the probes they used in Figure 6 – with panels of several nuclei (in supplementary data if necessary) in order for the reader to assess the efficiency of FISH detection. Currently just one nucleus is shown for the whole paper. In particular, given that the authors do not find the same results as another study from the Meister lab for three endogenous *rex* sites, it is important that actual data are shown.

5) In Figure 1—figure supplement 1 the authors show that expression of endogenous genes on the X does not differ from that of autosomal genes. The authors use this data to argue that autosomes are variable and thus that the X cannot be evaluated. However, an alternative interpretation is that the X cannot be shown to differ from autosomes, as previously reported (Gupta et al., 2006: Deng et al., 2011). This should be mentioned in the text.

6) As mentioned in the general comments, the Abstract and the rest of the text need to be modified to take into account the caveats discussed above and remove some statements. For example, the authors claim to provide "robust evidence against Ohno's hypothesis", "compelling evidence against upregulation", "refuting an X-chromosome-wide mechanism to balance gene expression between X and autosomes (Ohno's hypothesis)" – the data may argue against the hypothesis but the hypothesis cannot be ruled out so far based on the study of transgenes.

The authors argue that their conclusion is robust (subsection “Transgenes on X are expressed at half the level as transgenes on autosomes in XX animals, contrary to Ohno's hypothesis of a chromosome-wide mechanism to upregulate X expression”, last paragraph) for "genes on X". Yet, what was assayed were "transgenes inserted on X".

"[…] we disprove an X-chromosome dosage compensation model contingent upon DCC-dependent positioning of X relative to the nuclear periphery." The authors cannot rule this model out based on their own experiments, even if they do not find that their tgs with/without *rex* sites go to the nuclear envelope.

The second-to-last sentence of the Abstract says "[…] expression of X-linked transgenes is half that of autosomal transgenes […]" What the authors show is that in XX hermaphrodites, two X-linked transgene copies are expressed at half the level of two autosomal transgene copies, and in XO males, one X-linked copy is expressed at the same level as one autosomal copy. This sentence should be changed to reflect this specific information about the sex and copy number of these X-linked-to-autosomal transgene comparisons.

7) Further points that should be considered for the Discussion:

As mentioned by the authors, there is ample prior evidence that expression levels of dosage-sensitive genes involved in protein complexes are critical for fitness and thus it is likely that only such genes would be regulated (Pessia et al., 2012). The authors should expand these points more in the discussion. The transgene system used in the current study is not under any pressure to adjust expression. This system is artificial and in fact results in additional copies of genes, which does not seem to affect fitness. This should be considered in the Discussion.

In the Discussion, evolutionary comparisons by Julien et al. (2012) are considered as the definite study to refute the existence of upregulation of X-linked genes in mammals; yet, this study compares a small subset of X-linked and autosomal genes expressed in very different organisms and the question of whether chicken hemizygous for the chromosomes homologous to the mammalian X would survive has not been addressed. Interestingly, the authors of the present paper show that a previous evolutionary comparison that had concluded that there was no evolutionary upregulation in *C. elegans* (Albritton et al., 2014), was flawed due to the presence of germ cells at the stage of development studied. The current re-evaluation could actually support at least partial upregulation of the X. This may also be discussed.

[Editors' note: further revisions were requested prior to acceptance, as described below.]

Thank you for resubmitting your work entitled "Chromosome-wide mechanisms to decouple gene expression from gene dose during sex-chromosome evolution" for further consideration at *eLife*. Your revised article has been favorably evaluated by Kevin Struhl as the Senior editor, a Reviewing editor, and two reviewers.

The manuscript has been improved but there are some remaining issues that need to be addressed before acceptance, as outlined below. The paper will be accepted pending these changes:

*Reviewer #2:*

In this revised version the authors have responded to most comments of the reviewers. The few comments below should still be considered:

1) The authors should clearly define what they mean by "X chromosome-wide upregulation" and specify that X-inserted transgenes would only test the ability of this molecular mechanism to spread a short distance in cis. This should be discussed.

2) In the subsection “Transgenes on X are expressed at half the level as transgenes on autosomes in XX animals, contrary to a chromosome-wide mechanism to upregulate X expression”, three sentences in the first paragraph should be edited to match the figure legend and the rest of the text:

“[…] whether a chromosome-wide mechanism of upregulation functions in *C. elegans* to balance expression between X chromosomes and autosomes” [delete].

“[…] If an X-linked transgene is controlled by both a dosage compensation mechanism, which halves X expression in XX animals, and an upregulation mechanism, which doubles X expression in both sexes [delete], the per-copy expression [...]”

[...] Lastly, if the X chromosome is controlled by an Ohno-like model of chromosome-wide upregulation mechanism, an X-linked transgene in a dosage-compensation-defective XX mutant [...]

3) In the first paragraph of the subsection “Possible mechanisms to compensate for gene dose reduction during sex-chromosome evolution”, adding the following sentence would help clarify the discussion:

“[…] tolerated the evolution of sex chromosomes without a global compensation mechanism to correct for reduced X-chromosome gene expression. However, our results do not exclude the possibility that diverse gene-specific mechanisms might have arisen to elevate expression of individual X-linked genes with reduced dose (see below for a discussion of such mechanisms).”

4) In the fifth paragraph of the subsection “Possible mechanisms to compensate for gene dose reduction during sex-chromosome evolution”, note that additional mechanisms have been reported:

“[…] MOF, which acetylates histone H4 on lysine 16 to upregulate expression of a small set of genes on X (Deng et al., 2013). In addition, increases in RNA half-life and in the number of associated ribosomes have been reported for mammalian X-linked genes (Yin et al., 2009; Deng et al., 2013; Faucillion and Larsson et al., 2015).”

*Reviewer #3:*

The text of the manuscript has been appropriately revised with respects to its conclusions regarding Ohno's hypothesis, specifically stating that the authors provide evidence against a chromosome-wide mechanism of X upregulation, not Ohno's hypothesis itself. This is a reasonably justified claim.

In the rebuttal letter and accompanying addition to the manuscript, the authors provide adequate justification of their choice of different genes and reporters to be used in the transgenes, namely that they sought to test transgenes with a diverse set of expression levels and tissue specificities.

---

## [Author Response]

*This manuscript addresses dosage compensation strategies between XX females and XO males in* C. elegans *using transgenes. Two types of imbalance occur due to the sex chromosomes in* C. elegans*: one between the sexes and the other between the single X copy in males and the autosomes present in two copies. Dosage compensation between the sexes is achieved by repression of each X chromosome in XX hermaphrodites by the well-studied DCC (dosage compensation complex) recruited to rex sites on the X chromosomes. Dosage compensation between autosomes and X chromosome is less well characterized. Using a large panel of single copy transgenes (two different reporters under three different gene promoters in various combinations located) on the X chromosome (28 transgenes) or on autosomes (36 transgenes), the authors investigate transcriptomes from XX and XO individuals for evidence of either type of dosage compensation. RNA Seq is used to assess the extent to which transgenes shows differences in expression when X-linked or autosomal, in XX versus XO individuals. L1 larvae taken within 3 hours of hatching were examined, as these do not yet have germ cells, thus avoiding the problem of X-chromosome silencing which would interfere with the accuracy of X/A expression measurements.*

We would like to clarify that we used combinations of 4 different reporters and 4 different promoters. Also, we analyzed gene expression of the transgenes using quantitative PCR. Data obtained by RNA-seq and GRO-seq were only used in Figure 1—figure supplement 1 to demonstrate different levels of expression for the 6 different *C. elegans* chromosomes.

*First, the authors demonstrate that the DCC can spread to all transgenes inserted on the X. This result is validated using homozygous DCC-deficient XX animals. Intriguingly the presence of a dosage complex binding site (DCC), whether a dox or a rex site, is not required for dosage compensation for tgs on the X chromosome and has no impact on expression of most tgs on autosomes.*

*Second, the authors examine the extent to which X-chromosome transgene expression is balanced between the X and autosomes (the Ohno hypothesis). The authors find that most X-linked reporter transgenes examined are expressed at half the level of autosomes. The authors propose that this goes against Ohno's hypothesis for X-chromosome-wide mechanism to balance gene expression between the X and autosomes.*

Third, the authors examine a recently proposed model that X-chromosome dosage compensation is based on rex-dependent nuclear positioning of the X chromosome. Using their reporter transgenes, the authors find no evidence that localization at the nuclear periphery in XO males is linked to increased X-chromosome expression, versus decreased binding of DCC at rex sites and internal relocalisation in XX animals.

We would like to clarify that we found no preferential localization of ectopic autosomal *rex* sites or endogenous X-linked *rex* sites to the nuclear periphery in males compared to hermaphrodites.

*Overall, the manuscript is well written and communicates the experiments and results clearly. The main issue concerns the interpretation of some of the data particularly for the second and third sections. To summarise, the authors provide good evidence to support their first claim, i.e., that reduction of X-linked gene expression operates mechanistically on a chromosome-wide basis. The single copy transgene data is compelling and confirms previous genome wide studies that demonstrated that DCC sites near endogenous X-linked genes are neither necessary nor sufficient for dosage compensation between the sexes.*

We appreciate the reviewers' enthusiasm for this set of experiments.

*The authors' second claim that their evidence argues against Ohno's hypothesis of X upreglation needs some clarification and rewording to make it clear that the experiments address the specific molecular model of a chromosome-wide mechanism of upregulation, which like dosage compensation, can extend to transgenes. As the authors have used transgenes, one interpretation of their data is that whatever the mechanism of up-regulation of the X relative to autosomes, this does not spread to transgenes inserted on the X. In fact, as different strategies of up-regulation may apply to different X-linked genes, it is difficult to probe this using transgenes. Furthermore, results from transgenes cannot really be used to deduce whether endogenous X-linked genes are up-regulated during evolution as the authors propose. Ohno's original text and hypothesis does not state that X up-regulation occurred on a chromosome-wide basis. Ohno hypothesized an up-regulation of X-linked genes during millions of years of sex chromosome evolution prior to the evolution of repression of X-linked gene expression in females. Inserting a transgene on the autosomes or X chromosome and requiring an immediate up-regulation of the X-linked copy in essentially one generation cannot be a valid test of this hypothesis.*

In his 1967 article, Ohno stated that: "During the course of evolution, an ancestor to placental mammals must have escaped a peril resulting from the hemizygous existence of *all* the X-linked genes in the male by doubling the rate of product output of *each* X-linked gene." Ohno did not require one single chromosome-wide mechanism, but he did state that "each" gene must have its products doubled.

We have rewritten the sections relevant to Ohno's hypothesis to reflect that we tested one plausible molecular mechanism (chromosome-wide) by which Ohno's hypothesis might function. As one example of our changes, we say the following at the end of the Introduction:

"While our transgene approach demonstrates a robust chromosome-wide mechanism to balance X gene expression between the sexes, it provides strong evidence against an analogous, chromosome-wide mechanism that would fulfill Ohno's hypothesis for balancing gene expression between X chromosomes and autosomes. […] Our results suggest that if upregulation did occur to compensate for gradual loss of genes during X-chromosome evolution, it proceeded by the emergence of diverse gene-specific mechanisms that would compensate for their reduced dose."

As you will see below, our original manuscript and our current manuscript devotes the entire last section of the Discussion to alternatives to a chromosome-wide mechanism to balance expression between X chromosomes and autosomes. The Discussion header was and is entitled, "Possible mechanisms to compensate for gene dose reduction during sex-chromosome evolution". We respond more extensively to the reviewers' comments above in the answers to specific points below.

Finally, the third set of conclusions, concerning the role of nuclear positioning in X regulation is subject to the same caveats as above: the behavior of transgenes may not reflect endogenous X genes. Therefore, the authors should explain better why the use of transgenes is an appropriate means to test the model. The authors do look at three endogenous rex sites, however, and do not come to the same conclusions as a previous study by the Meister lab, although the reasons for this difference are not clear and require further investigation.

Because our X-linked transgenes are fully responsive to the dosage compensation process, they are valid tools for assessing the nuclear positioning model. Regulation of X-linked transgene expression should fulfill expectations of a dosage compensation model if the model is correct. Instead, our gene expression results are inconsistent with expectations of this model. One does not need to invoke the absence of an X-chromosome-wide upregulation model to interpret these results as being inconsistent with the Meister model.

With regard to gene expression, the nuclear positioning model predicts that transgenes on X (which were shown to be dosage compensated in hermaphrodites) should have elevated expression in males relative to transgenes on autosomes due to a *rex*-dependent association of X with the nuclear periphery. However, we found that in males, expression of single-copy transgenes on the sole X chromosome was not different from expression of single-copy transgenes inserted on only one of two homologous autosomes.

Second, the nuclear positioning model predicts that in XX animals defective in DCC binding, expression of a transgene co-inserted with a closely linked *rex* site should be higher than expression of the same transgene inserted at the same locus without a *rex* site. However, we found that in XX animals, depletion of the DCC did not significantly increase the expression of any of the 12 *rex*-linked reporter transgenes on X chromosomes or autosomes compared to their expression at the same locations without a *rex* site. Thus, proximity to a *rex* site is not sufficient to cause elevation of transgene expression when the DCC is not bound to the site, and DCC binding adjacent to a transgene is not sufficient to repress its expression, either when it is on X or an autosome, unlike expectations from the nuclear positioning model.

These are direct experiments and direct results using dosage compensated transgenes on X that should have responded in a manner consistent with the Meister lab model, if their dosage compensation model were correct. Results like ours would most likely have precluded publication of the model had they been presented with the model. Instead, the model was based on no gene expression studies or DCC binding studies. Lack of an X-chromosome-wide upregulation mechanism is not necessary to interpret these results as being inconsistent with the Meister model. For continuity of the paper, it seemed like an appealing way to write the story, but obviously it caused confusion about exactly what our experiments mean. We have rewritten the sections to avoid this confusion.

These gene expression results plus the fact that we tested 5 endogenous *rex* sites on X and two on autosomes and found no preferential localization of the sites with the nuclear periphery in males compared to hermaphrodites, makes us confident that this particular dosage compensation model of *rex*-dependent nuclear localization is not correct.

This topic is addressed again below in our comments to specific points.

*Thus, although this work provides novel insight about the mechanism of X-linked gene regulation in C. elegans, it cannot completely test Ohno's hypothesis of evolutionary X up-regulation. The manuscript (title, Abstract, text, and figures) will need to be revised accordingly and a number of specific points that should also be addressed, as listed below.*

We have rewritten the Abstract, text, and figure legends to make it very clear that we have tested one plausible molecular mechanism, a chromosome-wide mechanism, by which Ohno's hypothesis might operate, if it functions. In every conclusion we state our results are contrary to a chromosome-wide mechanism of upregulation. We do not say they are contrary to "Ohno's hypothesis". The title only states the topic of the paper; it does not state the conclusions. Besides, dosage compensation is a chromosome-wide mechanism to decouple gene expression from gene dose during sex chromosome evolution. We do not understand the necessity to revise it. Please see answers to specific points below.

*Specific points to be addressed:*

*1) Although the use of L1 larvae is clearly very elegant strategy to avoid the inclusion of germ cells which have a silent X, and to avoid potential variation in expression linked to distinct developmental lineage, they also introduce a potential caveat, which is that at such an early stage, X/A gene expression balance may not in fact be in place. The authors cannot rule out that at later stages there is global or local, X-linked up regulation to balance expression with autosomes. The authors should discuss this. One way to confirm this would be to use older animals from which the germ cells have been mechanically removed or genetically depleted.*

The embryo and L1 stages of *C. elegans* are the stages most sensitive to disruptions in gene expression, making it highly likely that any chromosome-wide mechanism should influence expression in these young animals. For example, mutations in dosage compensation genes cause either embryonic or L1 lethality. Mutations in the majority of essential genes cause embryonic or L1 lethality. While our experiments don't rule out the possibility that an upregulation mechanism might become more potent or act, in part, after the L1 stage, we would be very surprised not to find evidence of the effects at the L1 stage. Furthermore, the X/A gene expression ratio is not widely variable over developmental stages if one uses germlineless animals to make the comparisons. Deng et al. 2011 showed the X:A expression ratios to be 1.08 (early embryo), 1.32 (late embryo), 1.21 (L1), 1.3 (L2), 1.2 (L4). Also, if one compares average expression of individual chromosomes from germlineless L4 larvae, the expression differences between X and autosomes is as variable as we report in embryos. We feel that any discussion of later timing for an upregulation phenomenon would require all of these arguments and the inclusion of gene expression data mentioned here. That discussion would become a distraction from our main points. Experiments examining animals older than the L1 stage would take a very long time for little gain. For these reasons, we prefer not to address the topic.

*2) The authors do not adequately explain their choice of genes and promoters in the inserted transgenes. How and why were the promoters for the GFP fusions chosen? Why is the promoter-driven GFP or tdTomato sometimes fused to histone H2B? Since all the conclusions of the manuscript are based on this transgene expression data, the authors should provide explicit justifications for each of the genes and promoters chosen. Also the authors should mention how tg copy number was verified to be single copy – how was this quantified?*

We had one basic motivation in using the variety of promoters and transgenes we used and that reason is best summarized in the sentence that was added: "Use of multiple promoters and reporters with different expression levels and tissue specificities allowed us to test diverse gene regulatory scenarios for responsiveness to dosage compensation." (Results)

We already defined and explained all the promoters in the same section to which we added the sentence above. Also, we already stated in the summary to the section that all transgenes were responsive to the dosage compensation process, "regardless of their location on X and the origin of their promoter, whether from the *C. elegans* X chromosome (*eft-4), C. elegans* autosomes (*eft-3* and *dpy-30*), or a *C. briggsae* autosome (*Cbr-unc-119*)." (Results)

Regarding transgene copy number, the Materials and methods section on MosSci and miniMos has been changed to incorporate the following revision: "Strains were constructed and transgene copy number was analyzed as in (Frøkjær-Jensenet al., 2008; Frøkjær-Jensenet al., 2014). The hallmark of multi-copy transgenes and transgenes resulting from imprecise insertion events is the incorporation of DNA from the cloning vector backbone into the worm genome. Therefore to obtain strains with single-copy transgenes, we performed PCR with primers specific to the vector backbone to identify and eliminate strains that carried complex transgene insertion events."

*3) The authors make the general conclusion that DCC binding adjacent to a transgene inserted on autosomes is not sufficient to regulate its expression. However Figure 5 shows that this is the case for 7/8 transgenes, which means that this statement is valid for 87% of transgenes tested, not necessarily all. The statement should be changed to "most" not "all".*

We modified our statement in the text to say: "These results indicate that strong DCC binding adjacent to a gene is generally not sufficient to regulate its expression". (Results)

*4) For the DNA FISH data, it was unclear why the probe size used changed (Figure 7 legend), depending on the age of the embryos examined. Also, it would be desirable that the authors show examples of FISH data for the probes they used in Figure 6 – with panels of several nuclei (in supplementary data if necessary) in order for the reader to assess the efficiency of FISH detection. Currently just one nucleus is shown for the whole paper. In particular, given that the authors do not find the same results as another study from the Meister lab for three endogenous rex sites, it is important that actual data are shown.*

FISH probes were generally 30 kb, except for probes (5 kb) used to examine *rex-32, rex-23, rex-47*, and *rex-8* in older embryos. The reason for the difference is that we had conducted the experiments for *rex-32, rex-23, rex-47*, and *rex-8* in older embryos using 5 kb probes before the Meister paper was published. We switched to 30 kb probes for all new analyses because that probe size was used by the Meister lab. We wanted to be consistent in the way experiments were conducted between the two labs, since we already knew that we had gotten different results. In fact, both probe sizes work well for testing nuclear positioning and give the same answers. We always checked *rex* localization in young embryos of the age used by the Meister lab and also older embryos to be certain that embryo age did not affect our conclusion.

In studies of *rex-33*, we used two different 30 kb overlapping FISH probes, each labeled with a different fluorescent dye, and only performed quantification on FISH spots that had signals from both fosmids to unambiguously identify *bona fide* FISH signals.

For all FISH probes used we have now included examples of embryos in Figure 7—figure supplement 2 and Figure 7—figure supplement 3.

*5) In Figure 1—figure supplement 1 the authors show that expression of endogenous genes on the X does not differ from that of autosomal genes. The authors use this data to argue that autosomes are variable and thus that the X cannot be evaluated. However, an alternative interpretation is that the X cannot be shown to differ from autosomes, as previously reported (Gupta et al., 2006: Deng et al., 2011). This should be mentioned in the text.*

With all due respect, we do not understand this comment. In Figure 1—figure supplement 1, we show that average expression of endogenous genes on X differs from the average expression of genes on individual chromosomes. We do not show that "expression of endogenous genes on the X does not differ from that of autosomal genes". We only show that the average expression of X is not different from the average expression of all autosomes together. Because the average expression of X is so different from the average expression of individual autosomes, we argued that average autosomal expression is not a proxy for the expression of the ancestral autosome.

*6) As mentioned in the general comments, the Abstract and the rest of the text need to be modified to take into account the caveats discussed above and remove some statements. For example, the authors claim to provide "robust evidence against Ohno's hypothesis", "compelling evidence against upregulation", "refuting an X-chromosome-wide mechanism to balance gene expression between X and autosomes (Ohno's hypothesis)" – the data may argue against the hypothesis but the hypothesis cannot be ruled out so far based on the study of transgenes.*

As mentioned above, we have changed the wording extensively in appropriate places.

*The authors argue that their conclusion is robust (subsection “Transgenes on X are expressed at half the level as transgenes on autosomes in XX animals, contrary to Ohno's hypothesis of a chromosome-wide mechanism to upregulate X expression”, last paragraph) for "genes on X". Yet, what was assayed were "transgenes inserted on X".*

We added the word "transgenes".

*"[…] we disprove an X-chromosome dosage compensation model contingent upon DCC-dependent positioning of X relative to the nuclear periphery." The authors cannot rule this model out based on their own experiments, even if they do not find that their tgs with/without rex sites go to the nuclear envelope.*

As mentioned above, we respectfully disagree with the reviewers' assessment of what our experiments mean, but we have modified the phrasing of the last sentence to "argue against" the model.

*The second-to-last sentence of the Abstract says "expression of X-linked transgenes is half that of autosomal transgenes […]" What the authors show is that in XX hermaphrodites, two X-linked transgene copies are expressed at half the level of two autosomal transgene copies, and in XO males, one X-linked copy is expressed at the same level as one autosomal copy. This sentence should be changed to reflect this specific information about the sex and copy number of these X-linked-to-autosomal transgene comparisons.*

That part of the sentence was changed to the following: "expression of compensated hermaphrodite X-linked transgenes is half that of autosomal transgenes." A 150-word Abstract does not permit enough space to say more than that, and this statement gets the main point across.

*7) Further points that should be considered for the Discussion:*

*There is ample prior evidence that expression levels of dosage-sensitive genes involved in protein complexes are critical for fitness and thus it is likely that only such genes would be regulated (Pessia et al., 2012). The transgene system used in the current study is not under any pressure to adjust expression. This system is artificial and in fact results in additional copies of genes, which does not seem to affect fitness. This should be considered in the Discussion.*

We, of course, agree with the reviewers, which is the reason why we devoted the entire last section of the Discussion in the original manuscript to this topic. The section is entitled, "Possible mechanisms to compensate for gene dose reduction during sex-chromosome evolution," and is 2.5 pages long. It ends with the statements:

"Some combination of mechanisms similar to those used in yeast and mammals might also be used in *C. elegans* to compensate for dose-sensitive genes on X. Indeed, if loss of genes from sex chromosomes were gradual, the genome would have had the opportunity to respond in a gene-by-gene fashion to compensate for the reduced X gene dose, thereby rendering a global mechanism of X-chromosome upregulation unnecessary."

Pessia et al., 2012 was prominently referenced in that original section, along with many other examples.

To emphasize this point further, we added the following sentence in the Introduction:

"Our results suggest that if upregulation did occur to compensate for gradual loss of genes during X-chromosome evolution, it proceeded by the emergence of diverse gene-specific mechanisms that would compensate for their reduced dose."

We also added a sentence in the Discussion:

"These results rule out one plausible molecular mechanism by which Ohno's hypothesis might work, suggesting that if upregulation does occur, it operates through multiple, diverse gene-specific mechanisms, as discussed later."

*In the Discussion, evolutionary comparisons by Julien et al. (2012) are considered as the definite study to refute the existence of upregulation of X-linked genes in mammals; yet, this study compares a small subset of X-linked and autosomal genes expressed in very different organisms and the question of whether chicken hemizygous for the chromosomes homologous to the mammalian X would survive has not been addressed. Interestingly, the authors of the present paper show that a previous evolutionary comparison that had concluded that there was no evolutionary upregulation in C. elegans (Albritton et al., 2014), was flawed due to the presence of germ cells at the stage of development studied. The current re-evaluation could actually support at least partial upregulation of the X. This may also be discussed.*

Regarding the Julien et al. (2012) paper, we added a qualifying phrase in the Discussion. We wish to point out that in the original Introduction we were careful to put the Julien et al. paper into context. The quote below from the original (and current) paper does exactly what the reviewer asked.

"Comparison between the extant mammalian X chromosome and the orthologous chicken autosome failed to reveal evidence for X-chromosome-wide upregulation in placental mammals (Julien *et al.*, 2012). In these species, genes on the single active X chromosome in males and females are expressed, on average, at half the level of the orthologous pair of autosomes, contrary to Ohno's hypothesis. Although the experimental approach failed to identify a chromosome-wide transcriptional mechanism of X upregulation, it left open the possibility that regulatory mechanisms might have arisen on a gene-by-gene basis to compensate for low activity of critical X-linked genes caused by chromosome-wide reduction of X expression."

Regarding the Albritton et al. (2014) paper, we appreciate the reviewers' assessment that the data might be used to argue in favor of at least partial upregulation of X. However, without knowledge of whether the autosomal *P. pacificus* genes are subject to germline silencing (as many autosomal genes are in *C. elegans*), the data cannot be used to conclude that X upregulation occurs. We added a sentence to that effect in the Discussion. (subsection “Evidence against an X-chromosome-wide mechanism of transcriptional regulation to balance gene expression between X chromosomes and autosomes”, fourth paragraph).

[Editors' note: further revisions were requested prior to acceptance, as described below.]

*The manuscript has been improved but there are some remaining issues that need to be addressed before acceptance, as outlined below. The paper will be accepted pending these changes:*

*Reviewer #2:*

*In this revised version the authors have responded to most comments of the reviewers. The few comments below should still be considered:*

*1) The authors should clearly define what they mean by "X chromosome-wide upregulation" and specify that X-inserted transgenes would only test the ability of this molecular mechanism to spread a short distance in cis. This should be discussed.*

We believe our paper illustrates the meaning of "chromosome-wide" by the experiments and results that address the dosage compensation status of transgenes. The meaning of "chromosome-wide" is spelled out in the Abstract by example before we comment on results from experiments to address upregulation of X.

Our Abstract says, "Using single-copy transgenes integrated throughout the *Caenorhabditis elegans* genome, we show that expression of all X-linked transgenes is balanced between XX hermaphrodites and XO males. Furthermore, proximity of a dosage compensation complex binding site is neither necessary to repress X-linked transgenes nor sufficient to repress transgenes on autosomes."

In other words, foreign genes integrated onto X become regulated by the same process that regulates endogenous X-linked genes. We interpreted these results in the Abstract to mean that "X is broadly permissive for dosage compensation and the DCC acts via a chromosome- wide mechanism to balance transcription between the sexes."

In the Abstract, we then compare the two processes (dosage compensation [i.e. down regulation] and X upregulation) directly and say "In contrast, no analogous X-chromosome-wide mechanism balances transcription between X and autosomes: expression of compensated hermaphrodite X-linked transgenes is half that of autosomal transgenes."

Given our wording in the Abstract we unfortunately do not see the way in which the meaning of "chromosome-wide" is unclear. "Chromosome-wide" is the opposite of "local" or "gene specific". We also make that point in the Introduction, by example, we say, "Our results suggest that if upregulation did occur to compensate for gradual loss of genes during X- chromosome evolution, it proceeded by the emergence of diverse gene-specific mechanisms that would compensate for their reduced dose."

Given the context of our statements regarding "chromosome-wide" we respectfully request that we be allowed to maintain our current wording in the text.

We are also perplexed by the reviewer's comment that our experiments "only test the ability of this molecular mechanism to spread a short distance in cis". We see no reason that the distance would be limited. The distance doesn't appear to be limited for dosage compensation. We therefore do not understand the reason to comment in the text on the reviewer's statement.

*2) In the subsection “Transgenes on X are expressed at half the level as transgenes on autosomes in XX animals, contrary to a chromosome-wide mechanism to upregulate X expression”, three sentences in the first paragraph should be edited to match the figure legend and the rest of the text:*

*“[…] whether a chromosome-wide mechanism of upregulation functions in C. elegans to balance expression between X chromosomes and autosomes” [delete].*

*[…] If an X-linked transgene is controlled by both a dosage compensation mechanism, which halves X expression in XX animals, and an upregulation mechanism, which doubles X expression in both sexes [delete], the per-copy expression [...]”*

[…] Lastly, if the X chromosome is controlled by an Ohno-like model of chromosome-wide upregulation mechanism, an X-linked transgene in a dosage-compensation-defective XX mutant […]

For all the reasons listed below, we respectfully request that we be allowed to keep our original wording in the text.

We think, but are not certain, that the reviewer suggests we delete references to "Ohno's hypothesis" in the subsection “Transgenes on X are expressed at half the level as transgenes on autosomes in XX animals, contrary to a chromosome-wide mechanism to upregulate X expression”. We feel that it is important to keep these references in the text so as to maintain clarity in the logic of our experiments. Our careful wording was chosen explicitly to prevent any reader from thinking that "a chromosome-wide" mechanism was essential to prove Ohno's hypothesis. We have asked 5 scientists not connected with this study (including 2 experts in fly dosage compensation) to read the sentences to see whether our wording is misleading. They understood our intended meaning, which we believe is exactly the intended meaning of the reviewer. They mentioned that our sentences would not be clear in the absence of references to Ohno's hypothesis.

In addition, our Introduction sets up our intention precisely, that we are testing one and only one model by which Ohno's hypothesis might operate:

"Moreover, monitoring expression of the same gene in the same species while varying only its location within the genome enabled a direct comparison of X and autosomal expression levels that tests one attractive molecular mechanism (a chromosome-wide mechanism) for upregulating X-chromosome transcription to balance gene expression between X and autosomes."

In contrast, the figure legend was written for the context of its figure, which specifically mentions Ohno's hypothesis. The legend is clear without mentioning those words.

We appreciate that the reviewer has read sections of our paper carefully, but we worry that certain sentences are being read out of context and perhaps given a meaning that is not in keeping with the deliberate message of our manuscript (from the Abstract to the last sentence), which we believe is in agreement with the reviewer's requests.

We note that our wording satisfies reviewer 3, who originally had concerns that we were overstating our conclusions about Ohno's hypothesis.

*3) In the first paragraph of the subsection “Possible mechanisms to compensate for gene dose reduction during sex-chromosome evolution”, adding the following sentence would help clarify the discussion:*

“[…] tolerated the evolution of sex chromosomes without a global compensation mechanism to correct for reduced X-chromosome gene expression. However, our results do not exclude the possibility that diverse gene-specific mechanisms might have arisen to elevate expression of individual X-linked genes with reduced dose (see below for a discussion of such mechanisms).”

Throughout the manuscript, we have assiduously mentioned the possibility of gene-specific mechanisms, including the example from the Introduction given above and the very last sentence of the paper.

We set up the Discussion to mention possibility of gene-specific mechanisms from the outset to make this point unambiguous from the very start. We said: "These results rule out one plausible molecular mechanism by which Ohno's hypothesis might work, suggesting that if upregulation does occur, it operates through multiple, diverse gene- specific mechanisms, as discussed later."

In addition, the legend to Figure 6 already contains virtually the exact sentence requested by the reviewer: "The results do not exclude the possibility that diverse gene-specific mechanisms might have arisen to elevate expression of individual X-linked genes with reduced dose."

Given that the Discussion was set up to obviate any ambiguity about our conclusions, we propose that the spirit of the reviewer's request can be met by changing the parenthetical phrase "(see below)" to the parenthetical sentence that follows, without repeating a sentence already present in the manuscript: "(See discussion below for alternative mechanisms of compensation)."

*4) In the fifth paragraph of the subsection “Possible mechanisms to compensate for gene dose reduction during sex-chromosome evolution”, note that additional mechanisms have been reported:*

*… MOF, which acetylates histone H4 on lysine 16 to upregulate expression of a small set of genes on X (Deng et al., 2013). In addition, increases in RNA half-life and in the number of associated ribosomes have been reported for mammalian X-linked genes (Yin et al., 2009; Deng et al., 2013; Faucillion and Larsson et al., 2015).*

In compliance with the reviewer's request we added the following sentence:

"In addition, for some X-linked genes, evidence also exists for enhanced mRNA stability or enhanced translational efficiency via increased ribosome density as possible mechanisms to compensate for reduced gene dose (Deng et al., 2013; Faucillion and Larsson, 2015)."

The reference to Yin et al., 2009 was not included for two reasons: (1) The Yin et al.2009 study did not address expression in males, making it unclear whether it fits into the regime of "Ohno's hypothesis". In comparison, the Faucillion and Larsson study addressed male gene expression explicitly. (2) Faucillion and Larsson raise other relevant caveats regarding the NMD study of Yin et al.2009 that suggest the effect they see on X might be indirect due to the mutation used in the study and not an explicit mechanism of compensation.